# Hydrology and riparian forests drive carbon and nitrogen supply and DOC:NO$_3^-$ stoichiometry along a headwater Mediterranean stream

José L. J. Ledesma[1,2], Anna Lupon[2], Eugènia Martí[2], Susana Bernal[2]

[1]Institute of Geography and Geoecology, Karlsruhe Institute of Technology (KIT), Karlsruhe, 76131, Germany
[2]Integrative Freshwater Ecology Group, Centre for Advanced Studies of Blanes, Spanish National Research Council (CEAB-CSIC), Blanes, 17300, Spain

*Correspondence to*: José L. J. Ledesma (jose.ledesma@kit.edu)

**Abstract.** In forest headwater streams, metabolic processes are predominately heterotrophic and depend on both the availability of carbon (C) and nitrogen (N) and a favourable C:N stoichiometry. In this context, hydrological conditions and the presence of riparian forests adjacent to streams can play an important, yet understudied role determining dissolved organic carbon (DOC) and nitrate (NO$_3^-$) concentrations and DOC:NO$_3^-$ molar ratios. Here, we aimed to investigate how the interplay between hydrological conditions and riparian forest coverage drives DOC and NO$_3^-$ supply and DOC:NO$_3^-$ stoichiometry in an oligotrophic headwater Mediterranean stream. We analysed DOC and NO$_3^-$ concentrations, and DOC:NO$_3^-$ molar ratios during both base flow and storm flow conditions at three stream locations along a longitudinal gradient of increased riparian forest coverage. Further, we performed an event analysis to examine the hydroclimatic conditions that favour the transfer of DOC and NO$_3^-$ from riparian soils to the stream during storms. Stream DOC and NO$_3^-$ concentrations were generally low (overall averages ± SD were 1.0 ± 0.6 mg C L$^{-1}$ and 0.20 ± 0.09 mg N L$^{-1}$), although significantly higher during storm flow compared to base flow conditions in all three stream sites. Optimal DOC:NO$_3^-$ stoichiometry for stream heterotrophic microorganisms (corresponding to DOC:NO$_3^-$ molar ratios between 4.8 and 11.7) was prevalent at the midstream and downstream sites under both flow conditions, whereas C-limited conditions were prevalent at the upstream site, which had no surrounding riparian forest. The hydroclimatic analysis of storms suggested that large and medium storm events display a distinct mechanism of DOC and NO$_3^-$ mobilization. In comparison to large storms, medium storm events showed limited hydrological responses that led to significantly lower increases in stream DOC and NO$_3^-$ concentrations. During large storm events, different patterns of DOC and NO$_3^-$ mobilization depending on antecedent soil moisture conditions arise: drier antecedent conditions promoted rapid elevations of the riparian groundwater table, hydrologically activating a wider and shallower soil layer, and leading to relatively higher increases in stream DOC and NO$_3^-$ concentrations compared to large storm events preceded by wet conditions. Our results suggest that (i) increased supply of limited resources during storms can potentially sustain in-stream heterotrophic activity during high flows, especially during large storm events preceded by dry conditions, and (ii) C-limited conditions upstream were overcome downstream, likely due to higher C inputs from riparian forests present at lower elevations. The contrasting spatiotemporal patterns in DOC and

NO$_3^-$ availability and DOC:NO$_3^-$ stoichiometry observed at the studied stream suggests that groundwater inputs from riparian forests are essential for maintaining in-stream heterotrophic activity in oligotrophic, forest headwater catchments.

## 1 Introduction

The metabolism of lotic systems is highly dependent on environmental conditions (Bernhardt et al., 2018). In forest headwater streams, metabolism is dominated by the activity of heterotrophs because light availability is limited for primary producers and allochthonous matter inputs constitute the major energy source (Tank et al., 2018). In these settings, the supply and availability of carbon (C) and nutrients, including nitrogen (N), can constrain both the activity and the composition of stream heterotrophic microorganisms (Dodds et al., 2002; Brookshire et al., 2005; Fasching et al., 2020).

Consequently, biotic assimilation of both dissolved organic carbon (DOC) and nitrate (NO$_3^-$) have been shown to relate closely with ecosystem respiration in streams worldwide (Pastor et al., 2014; Catalán et al., 2018; Tank et al., 2018; Lupon et al., 2020a), which highlights the strong linkages between resource supply, demand, and in-stream heterotrophic activity. Yet, even if DOC and NO$_3^-$ are available, DOC:NO$_3^-$ stoichiometry can further influence in-stream heterotrophic activity (Elser et al., 2000; Helton et al., 2015). Heterotrophic organisms can adapt to a wide range of resource stoichiometric ratios

(Manzoni et al., 2017), and thus it is difficult to establish general reference values defining optimal conditions for in-stream heterotrophic activity. For example, global studies on stoichiometric controls for a wide range of ecosystems show discrepant reference C:N molar ratios for streams and rivers; ranging from 4.8 (Taylor and Townsend, 2010) to 11.7 (Helton et al., 2015). Modelling approaches using bacterial cultures show that heterotrophic microorganisms can perform well under C:N molar ratios ranging between 5.6 and 18.4 (Danger et al., 2008). Empirical case studies in forest headwater streams

reported a C:N range of 5.9 to 13.4 for optimal microbial activity (Pastor et al., 2014). Hence, considering that ecological stoichiometry theory sets biochemical constraints for metabolic activity based on the comparison of element ratios between resources and organisms (Sterner and Elser, 2002), there is a wide range of potentially optimal stoichiometric conditions for microbial heterotrophs. Nevertheless, C:N molar ratios outside this range may lead to either C- or N-limitation for heterotrophic activity.

In this context, hydrological processes at the riparian-stream interface, and in particular the mobilization of solutes from riparian soils to streams, can play an important role in determining stream DOC and NO$_3^-$ availability. Riparian zones are acknowledged as the major source of DOC in a wide range of forest headwater streams because their soils generally exhibit high organic matter contents, especially in shallower layers (Dosskey & Bertsch, 1994; Bernal et al., 2005; Köhler et al., 2009; Strohmeier et al., 2013). Relatively wet conditions in the riparian zone favour the build-up of this organic matter and

can also lead to denitrification, limiting the mobilization of NO$_3^-$ (McClain et al., 1994; Cirmo and McDonnell, 1997). However, riparian zones that experience frequent dryness, such as those in Mediterranean catchments, can manifest nitrification and be net sources of NO$_3^-$ to streams (Lupon et al., 2016a). As a result of these riparian zone characteristics, increased DOC concentrations during storm events in forest headwaters is an almost universal phenomenon (Hinton et al.,

1997; Boy et al., 2008; Yang et al., 2015; Musolff et al., 2018), whereas stream $NO_3^-$ concentrations can increase, decrease, or remain unchanged during storms (Àvila et al., 1992; Bernal et al., 2002; Inamdar and Mitchell, 2006; Dupas et al., 2017).

At the same time, climate and associated soil moisture conditions preceding storm events can influence both the amount of DOC and $NO_3^-$ stored in riparian soils and the riparian soil layers that are hydrologically connected to the stream, strongly influencing the supply of C and N to the aquatic compartment (Àvila et al., 1992; Fovet et al., 2015; Zimmer and McGlynn, 2018; Werner et al., 2019; Wen et al., 2020). Resulting stream DOC and $NO_3^-$ concentrations and DOC:$NO_3^-$ stoichiometry might therefore significantly differ between base flow and storm flow conditions, as well as among storms depending on antecedent accumulated precipitation, and thereafter influence in-stream DOC and $NO_3^-$ availability for microbial communities. Hence, studying the role of hydrology in controlling DOC and $NO_3^-$ supply from riparian zones to adjacent streams becomes important for the subsequent understanding of in-stream heterotrophic activity.

In this study, we aimed to investigate how the interplay between hydrological processes, including those related to storm events, antecedent climate conditions, and riparian forest coverage drives DOC and $NO_3^-$ availability and DOC:$NO_3^-$ stoichiometry in a forest headwater Mediterranean stream. In this stream, $NO_3^-$ is the major source of N for stream heterotrophic microorganisms, accounting for more than 90% of the dissolved inorganic N (Bernal et al., 2015) and for more than 80% of the total dissolved N (unpublished data). Yet, ambient DOC and $NO_3^-$ concentrations are relatively low compared to forest headwater streams located elsewhere (Hartmann et al., 2014; Bernal et al., 2015; Bernal et al., 2018), and we expected hydrological conditions and the presence of riparian forests to have a significant influence on the supply of these limited solutes for stream heterotrophic microorganisms. In order to fulfil our aim, we analysed DOC and $NO_3^-$ concentrations, and DOC:$NO_3^-$ molar ratios under base flow and storm flow conditions at three stream locations along a longitudinal gradient of increasing riparian forest coverage. We define riparian zones here as near-stream areas within the catchment where riparian forest is present. In addition, we performed an event analysis to examine the hydroclimatic conditions that most favoured the transfer of DOC and $NO_3^-$ from the riparian zone to the stream during storms, which were assumed to have a disproportionate influence on solute supply. Based on this hydroclimatic event analysis, we propose a conceptual model synthesising the complex hydroclimatic and biogeochemical processes driving DOC and $NO_3^-$ mobilization from riparian zones to streams during storm events.

## 2 Materials and methods

### 2.1 Study site characterization

The Font del Regàs catchment, located in the Montseny Natural Park in NE Spain (outlet at 41°50' N, 2°28' E), includes a total drainage area of 15.5 $km^2$ that ranges between 405 and 1603 m above sea level (m a.s.l) (Fig. 1). The climate is subhumid Mediterranean with mild winters, wet springs, and dry summers. Long-term (1940 to 2000) annual precipitation averaged $925 \pm 151$ mm $year^{-1}$ (average $\pm$ SD), whereas average annual air temperature during this period was $12.1 \pm 2.5$ °C (Lupon et al., 2016b).

The altitude gradient in the catchment drives changes in vegetation cover. While deciduous European beech (*Fagus sylvatica*) forests and a small proportion of heathlands (*Calluna vulgaris* and gramineae) dominate in the upper, steeper parts of the catchment, evergreen oak (*Quercus ilex*) forests dominate in the lower parts. Riparian forests, composed of a mixture of tree species including black alder (*Alnus glutinosa*), black locust (*Robinia pseudoacacia*), European ash (*Fraxinus excelsior*), and black poplar (*Populus nigra*), develop in flat, near-stream zones of the lower parts of the catchment (Lupon et al., 2016b). Both the width of the riparian forests and the total basal area of riparian trees increase downstream (Bernal et al., 2015). For its most part, Font del Regàs can be considered a closed-canopy stream (*sensu* Tank et al., 2018).

For this study, we selected three stream locations along a longitudinal gradient of increased riparian forest coverage, which delineate three nested subcatchments (Fig. 1). The upstream subcatchment (746 to 1603 m a.s.l.) drains a steep area of 1.8 km$^2$ along the first 2.9 km of the Font del Regàs stream. The vegetation here is dominated by beech forests and heathlands (92%), with lower proportions of oak forests (8%), and no riparian forests (i.e., no riparian zones). The midstream subcatchment (local drainage area of 6.8 km$^2$, ranging from 566 to 746 m a.s.l.) drains a cumulated area of 8.5 km$^2$ and includes an additional 1.8-km stream section. Cumulated vegetation proportions here are 53% beech forests, 43% oak forests, and 4% mixed, 5 to 15 m wide riparian forests. The downstream subcatchment (local drainage area of 4.4 km$^2$, ranging from 503 to 566 m a.s.l.) drains a cumulated area of 13 km$^2$ and includes an additional 1.4-km stream section, for a total stream length of 6 km. Cumulated vegetation proportions at the downstream subcatchment are 46% beech forests, 48% oak forests, and 6% riparian forests, which at the lower parts are well-developed and can be up to 30 m wide.

## 2.2 Field data collection, sampling, and laboratory analyses

The present study is based on data collected during the two-year period 9[th] September 2010 to 31[st] August 2012. All data and analyses were integrated and carried out for daily resolutions, which was the resolution of the stream chemical data.

Cumulated precipitation was recorded at 15 min intervals using an automatic weather station located at the valley bottom of the catchment, near the downstream site (Fig. A1). At each of the three stream locations, referred to as sites hereafter, we measured stream water level at 15 min intervals using pressure transducers (HOBO U20-001-04) connected to autosamplers (Teledyne Isco Model 1612). Stream flow was measured every two weeks at each site by applying the salt dilution method (Gordon et al., 2004). Rating curves obtained from the relationships between stream flow and stream water level were used to construct daily time series of stream flow data at each site (R$^2$>0.97 in all cases; Lupon et al., 2016b). Stream flow was normalized by drainage area; thus, it is denoted by units of mm throughout (n = 723 days in all cases; Fig. A1).

Groundwater table was recorded at 15 min intervals using a water pressure transducer (Druck PDCR 1830) within a piezometer placed 2.5 m away from the stream channel and located in a riparian zone area ca. 500 m downstream the downstream site (Fig. 1), where stream flow normalized by drainage area is assumed to be equivalent. In a previous study, we showed that the dynamics of the pressure transducer dataset captured well the dynamics of manually-measured groundwater table variations in seven other piezometers located in the surrounding riparian area (Ledesma et al., 2021). Therefore, we are confident that the recorded pattern at the pressure transducer location was representative of the

groundwater table variation in the riparian zone soils in the lower parts of the catchment. The available daily groundwater
table records were 664 (sporadic device failure precluded a full time series for the study period).

Stream water samples were collected every day at noon at each of the three sites using autosamplers (Teledyne Isco Model 1612), which were installed ca. 1 m belowground to keep samples cool and to prevent biogeochemical transformations (Lupon et al., 2016b). Water samples were transported to the laboratory every 7 to 14 days, where they were filtered through pre-washed Whatman GF/F glass fibre filters (pore size = 0.7 µm) and analysed for DOC by catalytic oxidation using a Shimadzu total organic carbon (TOC) analyser, and for $NO_3^-$ by cadmium reduction using a Technicon autoanalyzer. Sporadic autosampler failure precluded a full time series of data for the study period. Total number of available daily DOC concentration values for upstream, midstream, and downstream sites were 537, 668, and 620, respectively. Analogously, total number of available daily $NO_3^-$ concentration values were 552, 676, and 613, respectively. Stream DOC and $NO_3^-$ concentrations were used to calculate the DOC:$NO_3^-$ molar ratio for each available pair. Total number of DOC:$NO_3^-$ molar ratio values were 537, 668, and 611 for upstream, midstream, and downstream sites, respectively (Fig. A1).

### 2.3 Data treatment and statistical analyses

At each site, we classified daily stream flows into base flow and storm flow. This separation can be done using a variety of approaches and time resolutions, which in most cases require certain degree of subjectivity (Hewlett and Hibbert, 1967; Buffam et al., 2001; Fovet et al., 2018; Caillon and Schelker, 2020). Here, we defined the beginning of storm flow as the day when cumulated precipitation amounted more than 5 mm and stream flow was more than 10% higher than the previous day. We chose this precipitation threshold because hydrological responses to precipitation inputs below 5 mm day$^{-1}$ are generally undetectable in Mediterranean catchments (Gallart et al., 2002). The end of storm flow was defined as the day when stream flow was less than 5% lower than the previous day. All stream flow values within storm flow days were classified as storm flow, whereas the remaining values were classified as base flow. From the available 723 days, the upstream, midstream, and downstream sites included 128, 129, and 151 storm flow days. The small discrepancies in the storm flow correspondence among sites were likely caused by (i) differences in response times, being the downstream site slower in returning to base flow conditions; (ii) sporadic activations of intermediate intermittent tributaries; and (iii) storm origin, being eastern storms that first enter from the downstream part of the catchment more common than western storms, which enter from the upper and middle parts of the catchment.

In order to investigate the influence of hydrological conditions on stream chemistry, for each site, we compared stream DOC and $NO_3^-$ concentrations and DOC:$NO_3^-$ molar ratios between base flow and storm flow conditions using non-parametric Wilcoxon rank-sum tests for non-normally distributed data (Zar, 2010). Moreover, in order to investigate the impact of riparian forest coverage on stream chemistry, we further performed pairwise Wilcoxon rank-sum tests comparing DOC and $NO_3^-$ concentrations, and DOC:$NO_3^-$ molar ratios between stream sites (i.e., upstream *versus* midstream, upstream *versus*

downstream, and midstream *versus* downstream). This latter analysis was done separately for base flow and storm flow conditions.

Based on previous studies assessing optimal DOC:NO$_3^-$ molar ratios for microbial activity in lotic ecosystems, we classified stream DOC:NO$_3^-$ molar ratios into 'optimal', 'C-limited', and 'N-limited' for stream heterotrophic microorganisms. We used the reference DOC:NO$_3^-$ molar ratios for streams and rivers presented in the global reviews of Taylor and Townsend

(2010) and Helton et al. (2015) as the lower (i.e., 4.8) and upper (i.e., 11.7) limits delimiting the range of 'optimal' stoichiometric conditions. Consequently, stoichiometric conditions were considered 'C-limited' when stream DOC:NO$_3^-$ molar ratios were below 4.8 and 'N-limited' when stream DOC:NO$_3^-$ molar ratios were above 11.7. We explored the effect of hydrology and riparian forest coverage on DOC:NO$_3^-$ stoichiometry by calculating the frequency of optimal, C-limited, and N-limited conditions at each site and flow condition. We tested whether the frequencies of optimal, C-limited, and N-

limited conditions for a given site were statistically different between base flow and storm flow by using a contingency analysis (Zar, 2010). Contingency analysis was also employed to test whether there were statistical differences in the frequency of optimal, C-limited, and N-limited conditions among sites. This latter analysis was done separately for base flow and storm flow conditions.

Finally, we computed simple linear regressions between stream DOC concentration, stream NO$_3^-$ concentration, and stream

DOC:NO$_3^-$ molar ratio in order to investigate whether DOC or NO$_3^-$ concentration controlled changes in stoichiometry. Linear regressions were computed separately for each site and flow condition. Statistical analyses were carried out in the software JMP® Pro version 14.0.0 and the significance level for all analyses was set at $p<0.01$.

### 2.4 Hydroclimatic analysis of storm events

We further aimed to explore the hydroclimatic characteristics that most effectively promoted the mobilization of DOC and

NO$_3^-$ from the riparian zone into the stream via groundwater table elevation and consequent hydrological activation of upper riparian layers during storm events, which are generally associated with the largest exports of solutes from headwater catchments. For that, we identified the storm events that occurred during the study period and characterized them using nine hydroclimatic descriptors. We then related those descriptors with the observed stream DOC and NO$_3^-$ concentrations using partial least square (PLS) regression models. This analysis was performed only for data from the downstream site, where the

riparian zone was well-developed. Moreover, this was the only site with associated groundwater table data, which was used in some of the hydroclimatic descriptors. A detailed description of the analysis follows.

We identified storm events based on three requirements: (i) a precipitation amount during the days included in the event of at least 25 mm (events below these threshold showed marginal groundwater table responses at the temporal resolution of the study), (ii) stream flow was classified as storm flow during the days of the event (not all events with precipitation amounts

higher than 25 mm generated storm flow at the temporal resolution of the study), and (iii) a complete stream chemistry data series associated with the event for the downstream site was available (i.e., no gaps in chemical data for the event dates were

allowed because we were ultimately interested in stream chemical responses). Each event included all days classified as storm flow for the corresponding time period, plus the last day classified as base flow prior the days classified as storm flow, which was considered as the starting date of the event (Blaurock et al., 2021). This strategy ensured that the relevant

magnitude and range of the hydroclimatic descriptors associated with each event were covered. Moreover, when there was a substantial decline in stream flow and riparian groundwater table followed by a subsequent increase in both variables without reaching base flow conditions, we split this bimodal event into two events. In this case, the day immediately before the second increase in stream flow and riparian groundwater table was considered as both the last day of event $i$ and the first day of event $i+1$ (this was the case for events #1 and #2, presented later).

Each of the identified storm events was characterized by the following hydroclimatic descriptors: *duration* (days); precipitation amount ($P$, mm); accumulated precipitation seven days before the event ($P$-$7$, mm), as a proxy for short-range antecedent soil moisture conditions; accumulated precipitation 30 days before the event ($P$-$30$, mm), as a proxy for long-range antecedent soil moisture conditions; average stream flow ($Q_{avg}$, mm); average groundwater table ($Gw_{avg}$, m); and groundwater table range ($\Delta Gw$, m), calculated as the absolute difference between the highest and the lowest groundwater

tables measured during the event. Moreover, the hydroclimatic characterization of each event included the slope of the linear relationship between daily riparian groundwater tables and daily stream flows (*slope*, m mm$^{-1}$), as a proxy for the rate at which upper riparian layers are hydrologically activated in relation to increments in stream flow (Rodhe, 1989). We anticipated that this slope would vary depending on antecedent soil moisture conditions, with higher values related to drier conditions, and that this variation would have implications for solute mobilization and stream chemistry. Finally, we also

calculated the average stream flow normalized to average groundwater table ($Q_{avg}/Gw_{avg}$, mm m$^{-1}$), which served as proxy for average hydraulic conductivity in the activated layers of the riparian profile, with higher values indicating larger lateral transmissivity from the profile.

The nine hydroclimatic descriptors were considered as predictors in a PLS regression model, and average DOC and NO$_3^-$ concentrations at the downstream site during each storm event were the response variables. A PLS regression model is

appropriate when (i) the number of predictors is large relative to the number of observations and (ii) there is multicollinearity among predictors (Wold et al., 2001). For each response variable, a PLS regression model returns a goodness of fit ($R^2Y$), which is the variation of the response variable explained by the predictors, and a predictive ability ($Q^2Y$), which is an indication on how well the model can predict new data ($Q^2Y>0.50$ are considered good models). In addition, the model returns a VIP (variable influence on projection), a cumulative measure associated with each predictor that informs about the

relative importance of the predictor in the overall model. Predictors with VIP>1 are considered statistically important for the model performance (Eriksson et al., 1999). The PLS regression analysis was carried out in the software SIMCA (©Umetrics AB) version 14.0.

Finally, for a subset of storms for which clear hydrological activation of upper riparian soil layers occurred, we examined the type of hysteresis loop (clockwise, anticlockwise, or linear) of the relationship between daily riparian groundwater tables

and daily stream flows. This relationship typically shows clockwise hysteresis loops, which imply that, for a given stream flow, the groundwater table is higher during the rising limb than during the falling limb, indicating that near-stream zones are the dominant water sources during the rising limb (Kendall et al., 1999; Frei et al., 2010). We compared the type of hysteresis loop among these storm events in order to determine potential differences in runoff generation processes and associated solute mobilization and DOC:$NO_3^-$ stoichiometry.

## 3 Results

### 3.1 Stream chemistry and changes along the stream for base flow and storm flow conditions

Including all sites and dates, stream DOC concentration averaged $1.0 \pm 0.6$ mg C $L^{-1}$ and ranged from 0.2 to 5.4 mg C $L^{-1}$. Stream DOC concentration was significantly higher during storm flow than during base flow conditions in the three sites (Wilcoxon test, p<0.01). On average, stream DOC concentration increased similarly from base flow to storm flow in all
sites: by 66% at the upstream site (from $0.8 \pm 0.4$ to $1.4 \pm 0.8$ mg C $L^{-1}$), by 60% at the midstream site (from $1.0 \pm 0.5$ to $1.6 \pm 0.8$ mg C $L^{-1}$), and by 55% at the downstream site (from $0.9 \pm 0.4$ to $1.4 \pm 0.8$ mg C $L^{-1}$) (Fig. 2a-b). The variability in stream DOC concentration was relatively similar during both flow conditions, as indicated by similar coefficient of variations (CV): 54% and 57% in the upstream site, 46% and 52% in the midstream site, and 40% and 60% in the downstream site, respectively for base flow and storm flow conditions. All pairwise comparisons between sites showed
statistically significant differences in stream DOC concentration during base flow conditions (Fig. 2a). On average, stream DOC concentration increased by 19% from the upstream to the midstream site and then decreased by 10% from the midstream to the downstream site (Table 1). This pattern was similar during storm flow conditions, i.e., a 15% increase from the upstream to the midstream site and a 13% decrease from the midstream to the downstream site were observed, but differences in DOC concentration were statistically significant only between the midstream and the downstream sites in this
case (Fig. 2b).

Stream $NO_3^-$ concentration ranged from 0.09 to 1.10 mg N $L^{-1}$ considering all sites and dates, and averaged $0.20 \pm 0.09$ mg N $L^{-1}$. Stream $NO_3^-$ concentration was significantly higher during storm flow than during base flow conditions in the three sites (Wilcoxon test, p<0.01). The relative increase in stream $NO_3^-$ concentration from base flow to storm flow was remarkably similar among sites, i.e., 41% at the upstream site (from $0.26 \pm 0.05$ to $0.36 \pm 0.17$ mg N $L^{-1}$), 40% at the midstream site
(from $0.15 \pm 0.03$ to $0.22 \pm 0.11$ mg N $L^{-1}$), and 42% at the downstream site (from $0.17 \pm 0.03$ to $0.24 \pm 0.10$ mg N $L^{-1}$) (Fig. 2c-d). Moreover, stream $NO_3^-$ concentration was consistently more variable during storm flow than during base flow, as indicated by increases in the CV: from 20% to 47%, from 21% to 52%, and from 18% to 42% in upstream, midstream, and downstream sites, respectively. All pairwise comparisons between sites showed statistically significant differences in stream $NO_3^-$ concentration (Fig. 2c-d). On average, stream $NO_3^-$ concentration decreased 40% from the upstream to the midstream

site and increased nearly 10% from the midstream to the downstream site (Table 1). This pattern held during both base flow and storm flow conditions.

Considering the three sites pooled together, stream DOC:$NO_3^-$ molar ratio varied from 1 to 28, with an average of 6.1 ± 3.3. On average, stream DOC:$NO_3^-$ molar ratio was 20% higher during storm flow compared to base flow conditions for the upstream (increasing from 3.8 ± 2.1 to 4.6 ± 2.0) and midstream (increasing from 7.5 ± 3.4 to 8.9 ± 3.6) sites (Wilcoxon test, p<0.01) (Fig. 2e-f). There was no statistical difference in stream DOC:$NO_3^-$ molar ratio between base flow and storm flow conditions at the downstream site (Wilcoxon test, p = 0.08), where the overall average was 6.3 ± 2.7. All pairwise comparisons between sites showed statistically significant differences in stream DOC:$NO_3^-$ molar ratio (Fig. 2e-f). These differences were consistent between base flow and storm flow conditions: stream DOC:$NO_3^-$ molar ratio almost doubled between upstream and midstream sites and then decreased ca. 20% from the midstream to the downstream site (Table 1).

Stream DOC:$NO_3^-$ molar ratios considered optimal for heterotrophic microorganisms (i.e., from 4.8 to 11.7) were not prevalent at the upstream site during either base flow or storm flow conditions (Fig. 3). Nevertheless, at this site, the frequency of optimal DOC:$NO_3^-$ molar ratios was statistically higher during storm flow (33% of the storm flow time) than during base flow (18% of the base flow time), as indicated by the contingency analysis (p<0.01). The rest of the time, stream DOC:$NO_3^-$ molar ratio was lower than 4.8, indicating that C-limited conditions for heterotrophic activity dominated in the upstream site. By contrast, optimal DOC:$NO_3^-$ molar ratios were largely prevalent (ranging from 69% to 74% of the time) at the midstream and downstream sites during both base flow and storm flow conditions (Fig. 3). According to the contingency analysis, there was a statistical difference in the frequency of DOC:$NO_3^-$ stoichiometric conditions between base flow and storm flow for the midstream site (p<0.01). This difference was driven by an increase in N-limited conditions from 11% of the time during base flow to 24% of the time during storm flow, which was accompanied by a decrease in C-limited conditions from 15% of the time during base flow to 3% of the time during storm flow. At the downstream site, the frequency of DOC:$NO_3^-$ stoichiometric conditions was similar between base flow and storm flow conditions (contingency analysis, p = 0.31), and C-limited conditions were more frequent than N-limited conditions (25% *versus* 6% of the time for the combined base flow and storm flow periods). As a result of these patterns, there were differences in the frequency of DOC:$NO_3^-$ stoichiometric conditions among sites during both base flow and storm flow (contingency analyses, p<0.01).

During base flow, no relationship between stream DOC and $NO_3^-$ concentrations was found, whereas they were positively related during storm flow conditions at the three sites ($R^2$>0.39, p<0.01) (Fig. 4a-b). Strong positive linear relationships were observed between stream DOC concentration and stream DOC:$NO_3^-$ molar ratio during base flow conditions at the three sites ($R^2$>0.80, p<0.01) (Fig. 4c). During storm flow conditions, these relationships weakened because data became more scattered as DOC concentrations increased, but were still statistically significant in all sites ($R^2$>0.22, p<0.01) (Fig. 4d). By contrast, the three sites showed no relationship between stream $NO_3^-$ concentration and stream DOC:$NO_3^-$ molar ratio (Fig. 4e-f).

## 3.2 Hydroclimatic analysis of storm events

A total of 16 storm events satisfied our selection requirements for further analysis and were described using the suggested hydroclimatic descriptors (Table 2). A two-component PLS regression model using these 16 storm events as observations and the nine hydroclimatic descriptors as predictors explained 72% (i.e., $R^2Y = 0.72$) of the variation in average stream DOC concentration and 55% (i.e., $R^2Y = 0.55$) of the variation in average stream $NO_3^-$ concentration. The $Q^2Y$ values, representing the ability of the model to predict new data, were 0.49 for DOC and 0.25 for $NO_3^-$, both below the 0.50 threshold for good models. Three out of the nine predictors reached VIP>1, indicating that they were important for explaining the variability in the response variables (Fig. A2a). Specifically, higher precipitation amount (higher $P$), larger groundwater table range (higher $\Delta Gw$), and shallower average groundwater table (lower $Gw_{avg}$) related to higher stream concentrations of both DOC and $NO_3^-$, which fell close together in the PLS ordination biplot (Fig. 5a).

Remarkably, we observed two clusters of observations falling in two opposite sides across the first component of the PLS biplot: a dense cluster of eleven storm events located in the right side, close to $Gw_{avg}$ and opposite $P$ and $\Delta Gw$, and a more scattered cluster of five storm events located in the left side, closer to $P$ and $\Delta Gw$ and far from $Gw_{avg}$ (Fig. 5a). Indeed, there were large hydroclimatic differences between these two subsets of events. The eleven storm events clustered in the right side were characterized by (i) lower $P$, (ii) lower average stream flow ($Q_{avg}$), (iii) deeper $Gw_{avg}$, and (iv) smaller $\Delta Gw$ (i.e., smaller thickness of the riparian layer that becomes hydrologically activated during the event) compared to the five storm events located in left side (Wilcoxon rank-sum test, p<0.01 in all cases; Fig. 6a-d). Given the large difference in $P$ and associated $\Delta Gw$ between the two subsets of events, we classified them as 'medium storms' ($P$ = 29 to 98 mm, $\Delta Gw$ = 2 to 9 cm, n = 11) and 'large storms' ($P$ = 67 to 174 mm, $\Delta Gw$ = 21 to 55 cm, n = 5). Additionally, both stream DOC and stream $NO_3^-$ concentrations were significantly lower during the medium storm events than during the large storm events (Wilcoxon rank-sum test, p<0.01 in both cases; Fig. 6e-f). As a result, stream DOC concentration increased (on average) with respect to the average concentration observed during base flow conditions by 133% during large storms and only by 20% during medium storms. Likewise, stream $NO_3^-$ concentration increased 110% during large storms and only 21% during medium storms compared to average base flow conditions.

Furthermore, the nine hydroclimatic descriptors showed small variability among the medium storm events, but varied widely among the five large storm events (Table 2; Figure 6). Focusing on the latter, total *duration* varied between 4 and 11 days and $P$ between 67 and 174 mm. Accumulated precipitation seven ($P$-7) and, especially, 30 ($P$-30) days before the event (as proxies for short- and long-range antecedent soil moisture conditions, respectively) showed wide ranges, i.e., $P$-7 varied between 11 and 182 mm and $P$-30 varied between 39 and 426 mm. $Q_{avg}$ ranged from 0.8 to 4.2 mm, whereas $Gw_{avg}$ varied more moderately, between 0.64 and 0.87 m below the soil surface. $\Delta Gw$ varied between 0.21 and 0.55 m. The slope of the linear relationship between daily riparian groundwater tables and daily stream flows (*slope*), as a proxy for the rate at which upper riparian layers are hydrologically activated in relation to increments in stream flow, ranged from 0.06 to 0.18 m mm$^{-1}$

and was negatively related with both *P-30* ($R^2 = 0.74$) and the average stream flow normalized to average groundwater table ($Q_{avg}/Gw_{avg}$) ($R^2 = 0.68$). Finally, $Q_{avg}/Gw_{avg}$, used as a proxy for average hydraulic conductivity in the activated layers of the riparian profile, ranged from 0.9 to 6.1 mm m$^{-1}$. In addition, during the five large storm events, stream DOC and NO$_3^-$ concentrations were 55% to 178% higher than during base flow conditions (Table 2) and stream DOC concentration varied more (CV = 21%) than stream NO$_3^-$ concentration (CV = 13%) among these events.

Given their variability in hydroclimatic descriptors and in stream DOC and NO$_3^-$ concentrations, we performed a second PLS regression analysis using only the subset of five large storms in order to gain further insights into the hydroclimatic characteristics that most effectively mobilize DOC and NO$_3^-$ during these larger events. In this case, a two-component PLS regression model explained 96% (i.e., $R^2Y = 0.96$) and 94% (i.e., $R^2Y = 0.94$) of the variation in average stream DOC and NO$_3^-$ concentrations, respectively. The relative ordination of the predictors was similar to the original PLS regression, but the ability of the model to predict new data was high, with $Q^2Y$ values of 0.88 and 0.82 for stream DOC and NO$_3^-$ concentrations, respectively. Moreover, the important (VIP>1) predictors for explaining the variability in the response variables were four in this case and they were all different from the three important predictors of the original PLS model (Fig. A2b). Specifically, lower *P-30*, lower $Q_{avg}$, higher *slope*, and lower $Q_{avg}/Gw_{avg}$ related to
 higher stream concentrations of both DOC and NO$_3^-$ (Fig. 5b). These four predictors showed the highest loadings in the first component of the model, which explained most of the variation in stream DOC ($R^2Y = 0.68$) and NO$_3^-$ ($R^2Y = 0.86$) concentrations. From the four important predictors, *P-30* showed the largest loading in the second component and provided a distinct and extra statistical explanation for the DOC model compared to the NO$_3^-$ model. Hence, this component explained 28% of the variation in average stream DOC concentration, but only 8% of the variation in average stream NO$_3^-$ concentration in the second PLS.

Out of the five large storm events, which were the ones for which clear hydrological activation of upper riparian soil layers occurred, four showed clockwise hysteresis loops in their riparian groundwater table – stream flow relationship, whereas event #5, with the largest *P-30*, displayed a slight anticlockwise loop (Fig. 7a). Stream DOC:NO$_3^-$ molar ratios at the downstream site during the five events generally fell within the suggested optimal stoichiometric range (Fig. 7b). Nevertheless, event #5 deviated again from the general pattern and showed lower stream DOC:NO$_3^-$ molar ratios than the other events, partially falling within the range defined as C-limited conditions for heterotrophic activity.

## 4 Discussion

### 4.1 The role of the riparian zone on determining DOC, NO$_3^-$, and DOC:NO$_3^-$ along the stream

Stream DOC concentration at the Font del Regàs catchment ($1.0 \pm 0.6$ mg C L$^{-1}$) was remarkably low compared with other forest headwaters in boreal (de Wit et al., 2016), temperate (Musolff et al., 2018), and tropical (Boy et al., 2008) regions, but was comparable to other Mediterranean sites (Casas-Ruiz et al., 2017; Catalán et al., 2018). On the other hand, stream NO$_3^-$

concentration ($0.20 \pm 0.09$ mg N $L^{-1}$) was somewhat higher than those typically measured in N-limited boreal streams (Blackburn et al., 2017), comparable to the low concentrations measured in tropical and Mediterranean forest headwaters (McDowell et al., 1992; Catalán et al., 2018), and notably lower than those measured in temperate sites (Musolff et al., 2017). Therefore, Font del Regàs can be considered an oligotrophic stream with relatively low N and, especially, C availability. These conditions could constrain stream productivity in general and the activity of stream heterotrophs in particular, given that this is a predominantly heterotrophic system where ecosystem respiration rates are between 10- and 100-fold higher than gross primary production rates (Lupon et al., 2016c).

Clear and consistent patterns were observed for DOC and $NO_3^-$ concentrations and DOC:$NO_3^-$ molar ratios along the Font del Regàs stream (Table 1, Fig. 2). DOC concentration was especially low at the upstream site and increased moderately (ca. 20%) until the midstream site. This increase could be attributed to changes in catchment topography from upstream to midstream and the associated development of riparian forests in near-stream areas (Jencso et al., 2009; Musolff et al., 2018). The upstream site drains a steep area where soils are well-drained, and no riparian forest can develop. As geomorphology becomes more favourable for the development of wetter soils in the near-stream zone, riparian forests can establish and support relatively larger mobilization of DOC from the riparian zone to the stream (Inamdar and Mitchell, 2006; Bernal et al., 2018). However, DOC concentration decreased ca. 10% from the midstream to the downstream site despite that the riparian forest is well-developed along this section. This result suggests that DOC was not transported conservatively along the stream and that in-stream DOC demand was likely higher than supply between the midstream and downstream sites, as suggested also by Lupon et al. (2020b). This indication can be further supported by a more consistent optimal DOC:$NO_3^-$ stoichiometry in this section compared to that observed between the upstream and midstream sites, where stoichiometric conditions evolved from C-limited to optimal.

Concentrations of $NO_3^-$ were highest at the upstream site, and then notably decreased (ca. 40%) in the stretch to the midstream site. This result is in line with the idea that headwater streams can remove substantial amounts of $NO_3^-$ within relatively short distances (Peterson et al., 2001). Nevertheless, previous studies in this stream suggest that in-stream processing alone might had not been enough to explain the observed decrease in $NO_3^-$ concentration along the upstream-midstream section (Bernal et al., 2015). Denitrification in riparian zones developed along this section could provide groundwater inputs with low $NO_3^-$ concentrations and contribute to the reduction in stream $NO_3^-$, a process widely observed in temperate forest catchments (Cirmo and McDonnell, 1997). However, riparian soils at Font del Regàs can act as sources of $NO_3^-$ to the stream because they are well-oxygenated and can sustain large nitrification rates (Lupon et al., 2016a). Thus, it is unclear why the upstream site showed such consistently higher $NO_3^-$ concentration with respect to the midstream site, but a combination of steeper slopes and vegetation dominated by beech forest could partially explain this pattern because these features may lead to further oxygenation and potential nitrification in soil water and to lower $NO_3^-$ uptake compared to the oak and riparian forests (Schiff et al., 2002; Poblador et al., 2019; Simon et al., 2021). The small increase (ca. 10%) in $NO_3^-$ concentration between the midstream and downstream sites could in this case be explained by the increasing influence

of $NO_3^-$-rich riparian groundwater inputs in the downstream areas, though in-stream mineralization of leaf litter inputs could also contribute to it (Bernal et al., 2015; Lupon et al., 2015).

As a result of the observed DOC and $NO_3^-$ patterns among the stream sites, DOC:$NO_3^-$ molar ratios doubled from the upstream to the midstream site. This change could have important implications for stream heterotrophic activity because stoichiometric conditions shifted from predominately C-limited at the upstream site to predominately optimal at the midstream site (Fig. 3). C-limited conditions at the upstream site were clearly driven by low DOC availability, which is in accordance with previous studies showing strong C-limitation in Mediterranean and semi-arid streams (Catalán et al., 2018).

Supporting this idea, we found that stream DOC concentration was responsible for driving stream DOC:$NO_3^-$ molar ratios at the three sites under almost all circumstances (Fig. 4). As the influence of the riparian forest in the catchment increased from upstream to midstream and downstream locations, potential C-limitation based on DOC:$NO_3^-$ molar ratios was generally overcome, and stoichiometric optimal conditions for stream heterotrophic activity became prevalent. This result suggests that riparian zones are essential ecosystem compartments for ensuring the supply of DOC to the stream and, thus, for fuelling in-

stream heterotrophic activity (Lupon et al., 2020b).

It is worth mentioning that defining an optimal range of DOC:$NO_3^-$ molar ratios for in-stream heterotrophic activity is difficult due to the lack of consistent reference values, and the fact that heterotrophic microorganisms can adapt to a wide range of resource stoichiometric ratios (Manzoni et al., 2017). We acknowledged this uncertainty by considering a wide range of DOC:$NO_3^-$ values (from 4.8 to 11.7) defining potential optimal conditions for in-stream heterotrophic activity based

on published data compilations (Taylor and Townsend, 2010; Helton et al., 2015). However, this range could be even wider if modelling and site-specific studies are considered (Danger et al., 2008; Pastor et al., 2014; Bastias et al., 2020). Noteworthy, we obtained very similar frequencies of optimal, C-limited, and N-limited conditions when using a less restrictive DOC:$NO_3^-$ molar ratio range for optimal conditions (from 4.5 to 16) (Fig. A3). Therefore, we are confident that the approach used in the present study is helpful for illustrating different stoichiometric conditions and how they relate to C

and N availability for stream heterotrophic microorganisms at the study site and other similar sites elsewhere.

### 4.2 Changes in DOC, $NO_3^-$, and DOC:$NO_3^-$ associated with hydrological conditions: potential implications for in-stream heterotrophic activity

Stream DOC and $NO_3^-$ concentrations substantially increased during storm flow conditions at the three sites, a pattern consistent with previous observations in other Mediterranean catchments and that can be explained by the hydrological

activation of soil layers with relatively higher organic matter content in near-stream zones (Àvila et al., 1992; Bernal et al., 2002). At the upstream site, the magnitude of change between flow conditions was different between DOC (which on average increased by 66%) and $NO_3^-$ (which on average increased by 41%), leading to an increase in the frequency of optimal DOC:$NO_3^-$ stoichiometric conditions for heterotrophic activity during storm flow. By contrast, at the midstream and downstream sites, the frequency of optimal DOC:$NO_3^-$ stoichiometry was high regardless of flow conditions; although, at the

midstream site, N-limited and C-limited conditions became more and less frequent during storm flow, respectively. Overall, these results suggest that (i) hydrological inputs during storms increase stream DOC and $NO_3^-$ availability by supplying allochthonous dissolved organic matter (DOM) and nutrients from near-stream zones, potentially alleviating resource limitation, and (ii) in-stream heterotrophic activity might be further favoured during high flows in sites with predominant C-limitation due to an increase in the frequency of optimal stoichiometric ratios.

Nevertheless, higher water velocities and lower water residence times during storm flows can limit the capacity of heterotrophic microorganisms to take up essential elements from the water column, especially in headwater streams with steeper slopes. Thus, the biogeochemical opportunity for in-stream processing (*sensu* Marcé et al., 2018) might be restricted to locations downstream, as proposed by the pulse-shunt concept (Raymond et al., 2016). However, there are multiple lines of evidence resulting from studies carried out in our site and in other forest catchments that suggest that high stream flows might indeed offer *windows of opportunity* for in-stream heterotrophic activity in headwater locations. For example, recent studies have reported that heterotrophic microorganisms can process some of the C and N entering the stream during high flow conditions in forest headwaters across ecoregions (Seybold and McGlynn, 2018; Wollheim et al., 2018). Bernal et al. (2019) showed that in-stream DOC and $NO_3^-$ uptake at Font del Regàs could be as high or even higher during storm flow than during base flow. Further, Bernal et al. (2018) showed that DOM at Font del Regàs has a prominent protein-like character in both riparian groundwater and stream water during base flow conditions, which could favour rapid assimilation even during periods of short water residence times (assuming that DOM molecular composition maintains part or most of its labile character across flow conditions). Sporadic inputs of DOM rich in aliphatic molecules have been observed in other forest headwaters during storm flow conditions, which could further increase bioavailability and assimilation (Wilson et al., 2013; Wagner et al., 2019). Part of the in-stream biogeochemical uptake during high flows could also be triggered by extra inflows of heterotrophic bacteria mobilized from catchment soils, as suggested by Caillon and Schelker (2020). Mechanistically, high flows can also enhance the hydrological interaction with streambed sediments, where microbial assemblages develop (Li et al. 2021). Lastly, and more importantly for the present study, changes in DOC, $NO_3^-$, and DOC:$NO_3^-$ along the Font del Regàs stream were remarkably similar during both base flow and storm flow conditions. This observation suggests that the pattern of resource consumption was maintained independently of stream flow conditions. All these direct and indirect pieces of evidence support that in-stream heterotrophic activity can be partially sustained during storm flows in forest headwater streams, especially in sites with limited C and N availability such as Font del Regàs.

**4.3 Storm size and antecedent soil moisture conditions shape the mobilization of DOC and $NO_3^-$**

Understanding which hydrological and climatic conditions result in more effective resource mobilization from riparian soils to the stream is relevant in the context of the present study and for other oligotrophic streams. Here, we focused this question on storm events, which are generally associated with the largest exports of solutes from headwater catchments and have the

capacity to hydrologically connect shallow, organic-rich riparian layers to the fluvial network (Godsey et al., 2009; Raymond et al., 2016; Zimmer and McGlynn, 2018).

Results from the first PLS regression model including all storm events showed that precipitation amounts and associated hydrological responses clearly distinguish two subsets of events. Large storms, characterized by higher precipitation amounts, produced higher stream flows and considerable groundwater table elevations, whereas medium storms produced only moderate responses in stream flow and limited groundwater table elevations (Table 2; Fig. 6). These differences in climatic and hydrological characteristics between large and medium storm events led to marked differences in the resulting stream chemistry, with both DOC and $NO_3^-$ concentrations being significantly higher during the large storm events. Hence, the first PLS showed that higher precipitation (higher $P$), larger groundwater table elevations (higher $\Delta Gw$), and shallower groundwater tables (lower $Gw_{avg}$) related to higher stream DOC and $NO_3^-$ concentrations (Fig. 5; Fig. A2). Mechanistically, chemical differences between the two types of events are likely explained by the hydrological activation of a thicker and shallower riparian layer during large storm events that leads to the mobilization of relatively larger amounts of DOC and $NO_3^-$ stored in the riparian soil compared to the amounts mobilized from the deeper and narrower layers that are hydrologically activated during medium storm events (Fig. 8a). These results suggest that large and medium storm events display distinct mechanisms of DOC and $NO_3^-$ mobilization from the riparian zone and that large storms are responsible for a disproportionally larger supply of these solutes.

Thus, storm size and associated groundwater table responses provide a first order control (in terms of relevance) on the mobilization of DOC and $NO_3^-$ from the riparian zone in the Font del Regàs catchment. Nevertheless, in our dataset of 16 storms, we observed an overlap in the 67 to 98 mm precipitation range in which two events (*6 and *7) showed small groundwater table elevations (4 and 9 cm, respectively) and were classified as medium storms, and two other events (#1 and #5) showed significant groundwater table elevations (36 and 21 cm, respectively) and were classified as large storms. Therefore, there is an apparent threshold range in which storm size (defined only by precipitation amount) can lead to either substantial or limited groundwater table responses and, thereby, to relatively more or less DOC and $NO_3^-$ mobilization. While the sample size (n = 4) is too small to clearly determine what processes or pre-conditions define whether a storm in the 67 to 98 mm range will lead to significant groundwater table responses, we noted that medium storm events *6 and *7 took place in July-August and October, respectively, whereas large storm events #1 and #5 took place in March and November, respectively. We argue that seasonality and, in particular, the differences in evapotranspiration between the vegetative and the dormant seasons, might provide a plausible explanation in this context. Mediterranean catchments such as Font del Regàs experience long periods of high evapotranspiration that lead to catchment drying and hydrological disconnection in late spring-summer and a subsequent re-wetting in autumn-late autumn (Medici et al., 2008). In this sense, the time of the year when a storm occurs might determine the magnitude of the riparian groundwater table response and the consequent DOC and $NO_3^-$ mobilization for intermediate-medium to large storm sizes in Mediterranean catchments. Future

studies should specifically investigate this ambiguous catchment response to similar precipitation inputs in light of the results presented here (e.g. Beiter et al., 2020).

The second PLS regression analysis focussed on understanding which specific hydrological and climatic conditions result in more effective resource mobilization from riparian soils only during large storm events. While the sample size of large storms was small, the nature of these events covered a wide range of hydroclimatic characteristic and provided useful insights on the processes that control DOC and $NO_3^-$ mobilization from riparian zones to streams in Mediterranean forest headwater catchments during these larger events. Particularly, we found that long-range antecedent soil moisture conditions

and the relationship between riparian groundwater table and stream flow are essential factors for understanding changes in stream DOC and $NO_3^-$ concentrations during large storms. Antecedent drier conditions (lower *P-30*) led to relatively higher stream DOC and $NO_3^-$ concentrations and constituted the most effective situation for the mobilization and supply of these solutes to the Font del Regàs stream. Therefore, large storms preceded by dry conditions might be essential for heterotrophic activity in this nutrient-limited stream. The large mobilization of solutes during these circumstances can be partially

explained by abrupt increases in riparian groundwater table, i.e., rapid hydrological activation of upper riparian layers in relation to increments in stream flow (higher *slope*). Hence, according to the second PLS analysis, drier antecedent conditions led to steeper slopes of the relationship between groundwater table and stream flow, which were related to lower average hydraulic conductivity in the activated layers of the riparian profile (lower $Q_{avg}/Gw_{avg}$), and to lower stream flows (lower $Q_{avg}$) (Fig. 5). A faster increment in riparian groundwater table position compared to the increment in stream flow can

be explained by a lower effective hydraulic conductivity of each riparian soil layer (estimated here as $Q_{avg}/Gw_{avg}$) induced by the pre-existing dry conditions and an increased hydrological disconnection between the riparian zone and the rest of the catchment. This preconditioning will make precipitation more efficient at filling up the riparian soil profile vertically relative to the otherwise dominant lateral transmissivity dimension (Frei et al., 2012; Fovet et al., 2015; Tunaley et al., 2016). Consequently, more riparian layers will be hydrologically connected to the stream, increasing the thickness of the 'Dominant

Source Layer' (i.e., the riparian zone depth stratum that contributes the most to water and solute fluxes to the stream, Ledesma et al., 2018) and, thus, the chances for relatively more DOC and $NO_3^-$ mobilization, which in turn generally show higher concentrations closer to the soil surface (Ranalli and Macalady, 2010; Camino-Serrano et al., 2016). In addition, drier conditions can restrict decomposition and promote primary production and nitrification in the riparian soil and result in the built up of soluble DOC and $NO_3^-$ pools (Lupon et al., 2016a; Arce et al., 2019; Wen et al., 2020), which will further

contribute to a higher mobilization of these solutes during subsequent large storm events, eventually leading to large increases in stream DOC and $NO_3^-$ concentrations (Fig. 8c). Conversely, the PLS analysis indicated that wetter antecedent soil moisture conditions and less steep slopes of the relationship between groundwater table and stream flow resulted in relatively lower increases in stream DOC and $NO_3^-$ concentrations during large storms. Under these circumstances, soil moisture will be high and lateral transmissivity will be relatively larger due to increased hydrological connectivity between

the riparian zone and the rest of the catchment. In addition, when large storms are preceded by wet conditions, soluble DOC

and $NO_3^-$ pools in the riparian soils can be depleted from precedent storm events (Butturini et al., 2003; Zimmer and McGlynn, 2018; Werner et al., 2019), eventually leading to relatively lower increments in stream DOC and $NO_3^-$ concentrations (Fig. 8b). Moreover, the relatively higher stream flows during storms preceded by wet conditions suggest larger spatial extension of water connectivity throughout the catchment (Tunaley et al., 2016), which may further contribute

to dilute solute inputs, especially in oligotrophic streams such as Font del Regàs.

Our conceptual model of riparian DOC and $NO_3^-$ mobilization during large storm events based on data from a sub-humid Mediterranean catchment is similar to previous conceptualizations presented for temperate sites. The model highlights that solute mobilization depends on antecedent soil moisture conditions and the relationship between riparian groundwater table and stream flow (Werner et al., 2019; Beiter et al., 2020). Thus, the mechanisms proposed here for large storm events can be

representative of other forest headwaters located in similar climatic settings. In addition, our analyses indicate that in Mediterranean forest headwaters, 'medium storms' show limited groundwater table responses and therefore solute mobilization is more restricted and less dependent on antecedent soil moisture conditions or the relationship between riparian groundwater table and stream flow. This distinctive feature of Mediterranean catchments compared to temperate sites might relate to their overall higher temperatures and evapotranspiration rates and, thus, could extent geographically in the future as

climate becomes warmer in temperate areas (Spinoni et al., 2018).

Importantly, our results additionally highlight that the relationship between riparian groundwater table and stream flow during large storms is not static but variable depending on storm event characteristics. This relationship typically shows clockwise hysteresis loops and synchronous behaviour patterns between riparian groundwater table and stream flow (Jung et al., 2004; Frei et al., 2010; Fovet et al., 2015). However, here we showed that clockwise hysteresis loops during large storms

were not consistently similar between events and that under very wet antecedent conditions the clockwise pattern can turn into anticlockwise, as seen for event #5 (Fig. 7a). Moreover, event #5 was the only large storm for which stream DOC:$NO_3^-$ molar ratios fell outside the optimal stoichiometric range for in-stream heterotrophic activity (Fig. 7b), which was explained by a relatively lower increase in stream DOC concentration compared to that in stream $NO_3^-$ concentration during the event (Table 2). This result suggests higher depletion and dilution of riparian DOC compared to riparian $NO_3^-$ during large storms

preceded by very wet conditions (Fig. 8b). This observation implies that C rather than N limitation might unfold in Mediterranean catchments such as Font del Regàs under those circumstances (Doblas-Miranda et al., 2013). Overall, these results underscore the relevance of climatic conditions preceding storm events, particularly large storm events, in controlling the relative availability of DOC and $NO_3^-$ in streams and, ultimately, the potential activity of heterotrophic microbial assemblages.

## 5 Conclusions

Spatiotemporal patterns of stream DOC and $NO_3^-$ concentrations and $DOC:NO_3^-$ molar ratios, and thus where and when C and N supply from terrestrial ecosystems can sustain or constrain in-stream heterotrophic activity, depend on catchment hydrological and biogeochemical processes, especially those occurring at the riparian zone. Supporting this idea, we found clear and consistent changes in DOC and $NO_3^-$ concentrations and $DOC:NO_3^-$ molar ratios during both base flow and storm flow conditions along a longitudinal gradient of increased riparian forest coverage at the Mediterranean Font del Regàs stream. This outcome indicates that catchment geomorphic and topographic features, which give raise to specific vegetation and hydromorphological features, determine stream DOC and $NO_3^-$ dynamics by modulating riparian forest formation (Inamdar and Mitchell, 2006; Ledesma et al., 2018).

Increases in both DOC and $NO_3^-$ concentrations during storm flow compared to base flow conditions is a common observation in forest headwaters and the Font del Regàs stream was not an exception. Nevertheless, in this study we went a step further by considering how changes in $DOC:NO_3^-$ stoichiometry along the stream and between flow conditions might impact in-stream heterotrophic activity. Optimal $DOC:NO_3^-$ ratios for heterotrophic microorganisms were prevalent at midstream and downstream locations regardless of hydrological conditions, whereas they were more frequent during storm flow than during base flow conditions upstream. We conclude that hydrology plays a critical role in controlling C and N supply from the riparian zone as well as $DOC:NO_3^-$ stoichiometric requirements in close-canopy, oligotrophic Mediterranean streams because solute mobilization during high flows can alleviate resource limitation and burst in-stream heterotrophic activity during or right after storm events.

Further, our results suggest that increases in stream DOC and $NO_3^-$ concentrations during storm events are controlled by complex hydrological processes involving storm size, seasonally-varying groundwater table responses, and antecedent soil moisture conditions. Particularly, large storms preceded by drier conditions lead to more rapid elevations of the riparian groundwater table and to larger increases in stream DOC and $NO_3^-$ concentrations, which contrasted to the small responses observed during medium storms. As the world climate continues changing, including an increased variability in precipitation patterns in the Mediterranean ecoregion (i.e., larger precipitation events and longer dry spells), it will be critical to understand hydrological and biogeochemical processes at the stream-riparian interface, and how the aftermath hydrology plays out for C and N availability, C:N stoichiometry, and consequent in-stream heterotrophic activity in Mediterranean forest headwaters.

**Data availability**

Precipitation, stream flow, groundwater table, and stream chemistry data used in the present study are available in the research data repository HydroShare at https://www.hydroshare.org/resource/3366012c5e254937aa661ce2a93c3140/ (Ledesma, 2021) and https://www.hydroshare.org/resource/0b4eb61ad7544ffb970c1600927a819f/.

**Author contribution**

JLJL: Conceptualization, Formal analysis, Funding acquisition, Methodology, Visualization, Writing – original draft preparation. AL: Conceptualization, Investigation, Resources, Writing – review & editing. EM: Conceptualization, Supervision, Writing – review & editing. SB: Conceptualization, Investigation, Resources, Supervision, Writing – review & editing.

**Competing interests**

The authors declare that they have no conflict of interest.

**Acknowledgements**

JLJL was funded by the project RIPARIONS granted by the European Commission through a Marie Skłodowska Curie Individual Fellowship (H2020-MSCA-IF-2018-834363) and by the Spanish Government through a Juan de la Cierva grant (FJCI-2017-32111). AL was supported by the Catalan Government and the European Commission through a MSCA-Beatriu de Pinós grant (BP-2018-00082). SB work was supported by the CANTERA project (RTI2018-094521-B-100) and a Ramón y Cajal fellowship (RYC-2017-22643) funded by MCIN/AEI/10.13039/501100011033/FEDER Una manera de hacer Europa.

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

**Table 1.** Relative differences in average dissolved organic carbon (DOC) and nitrate ($NO_3^-$) concentrations and DOC:$NO_3^-$ molar ratios between upstream and midstream, and between midstream and downstream sites at the Font del Regàs stream for the study period (September 2010 to August 2012). Positive values indicate that either solute concentrations or stoichiometric ratios increased in the downstream direction, while negative values indicate a decrease. Differences are shown separately for base flow and storm flow conditions.

| Section | Flow conditions | DOC difference | $NO_3^-$ difference | DOC:$NO_3^-$ difference |
|---|---|---|---|---|
| Upstream to midstream | Base flow | +19% | -40% | +98% |
| Upstream to midstream | Storm flow | +15% | -40% | +95% |
| Midstream to downstream | Base flow | -10% | +8% | -18% |
| Midstream to downstream | Storm flow | -13% | +9% | -23% |

**Table 2.** Hydroclimatic descriptors (n = 9) of the storm events identified during the study period (September 2010 to August 2012) at the downstream site of the Font del Regàs catchment, including duration, precipitation amount (P), accumulated precipitation seven (P-7) and 30 (P-30) days before the event, average stream flow ($Q_{avg}$), average groundwater table ($Gw_{avg}$), groundwater table range ($\Delta Gw$), slope of the linear relationship between riparian groundwater table and stream flow (slope), and average stream flow normalized to average groundwater table ($Q_{avg}/Gw_{avg}$). Average dissolved organic carbon (DOC) concentration and average nitrate ($NO_3^-$) concentration during each event at the downstream site and the relative increase in DOC and $NO_3^-$ concentrations with respect to average concentrations observed during base flow conditions (within brackets) are also shown. Events are subdivided into large (#) and medium (*) storms according to the results of a partial least square (PLS) regression analysis.

| Event | Period | Duration (days) | P (mm) | P-7 (mm) | P-30 (mm) | $Q_{avg}$ (mm) | $Gw_{avg}$ (m) | $\Delta Gw$ (m) | Slope (m mm$^{-1}$) | $Q_{avg}/Gw_{avg}$ (mm m$^{-1}$) | DOC (mg C L$^{-1}$) | $NO_3^-$ (mg N L$^{-1}$) |
|---|---|---|---|---|---|---|---|---|---|---|---|---|
| #1 | 11-14/03/2011 | 4 | 95 | 11 | 39 | 1.7 | 0.74 | 0.36 | 0.16 | 2.3 | 2.35 (167%) | 0.40 (140%) |
| #2 | 14-23/03/2011 | 10 | 116 | 102 | 133 | 3.3 | 0.64 | 0.55 | 0.14 | 5.1 | 2.16 (146%) | 0.33 (97%) |
| #3 | 02-12/11/2011 | 11 | 134 | 65 | 116 | 0.8 | 0.87 | 0.21 | 0.18 | 0.9 | 2.45 (178%) | 0.40 (138%) |
| #4 | 14-19/11/2011 | 6 | 174 | 14 | 261 | 4.2 | 0.69 | 0.52 | 0.06 | 6.1 | 1.91 (117%) | 0.29 (74%) |
| #5 | 21-30/11/2011 | 10 | 67 | 182 | 426 | 3.4 | 0.74 | 0.21 | 0.07 | 4.5 | 1.36 (55%) | 0.34 (102%) |
| *1 | 16-19/09/2010 | 4 | 52 | 9 | 36 | 0.4 | 0.97 | 0.04 | 0.10 | 0.4 | 1.01 (15%) | 0.23 (36%) |
| *2 | 14-17/05/2011 | 4 | 30 | 22 | 122 | 0.7 | 0.90 | 0.02 | 0.15 | 0.8 | 0.99 (13%) | 0.17 (5%) |
| *3 | 30/05-02/06/2011 | 4 | 36 | 0 | 69 | 0.6 | 0.92 | 0.02 | 0.15 | 0.6 | 1.19 (36%) | 0.18 (7%) |
| *4 | 03-06/06/2011 | 4 | 50 | 36 | 96 | 0.6 | 0.92 | 0.03 | 0.14 | 0.7 | 0.83 (-5%) | 0.17 (-1%) |
| *5 | 09-15/06/2011 | 7 | 34 | 66 | 148 | 0.6 | 0.91 | 0.03 | 0.15 | 0.7 | 1.10 (26%) | 0.18 (7%) |
| *6 | 25/07-03/08/2011 | 10 | 74 | 4 | 48 | 0.5 | 0.94 | 0.04 | 0.16 | 0.5 | 0.82 (-7%) | 0.18 (5%) |
| *7 | 23-30/10/2011 | 8 | 98 | 17 | 39 | 0.3 | 0.95 | 0.09 | 0.17 | 0.3 | 1.53 (75%) | 0.23 (41%) |
| *8 | 21-23/03/2012 | 3 | 57 | 6 | 8 | 0.8 | 0.88 | 0.06 | 0.22 | 0.9 | 0.95 (8%) | 0.36 (118%) |
| *9 | 03-05/04/2012 | 3 | 29 | 0 | 63 | 0.7 | 0.95 | 0.06 | 0.13 | 0.7 | 1.23 (40%) | 0.16 (-2%) |
| *10 | 18-24/05/2012 | 7 | 49 | 10 | 62 | 0.5 | 0.97 | 0.04 | 0.02 | 0.5 | 0.82 (-7%) | 0.19 (12%) |
| *11 | 04-09/08/2012 | 6 | 33 | 0 | 4 | 0.2 | 1.05 | 0.03 | 0.23 | 0.2 | 1.08 (23%) | 0.18 (5%) |

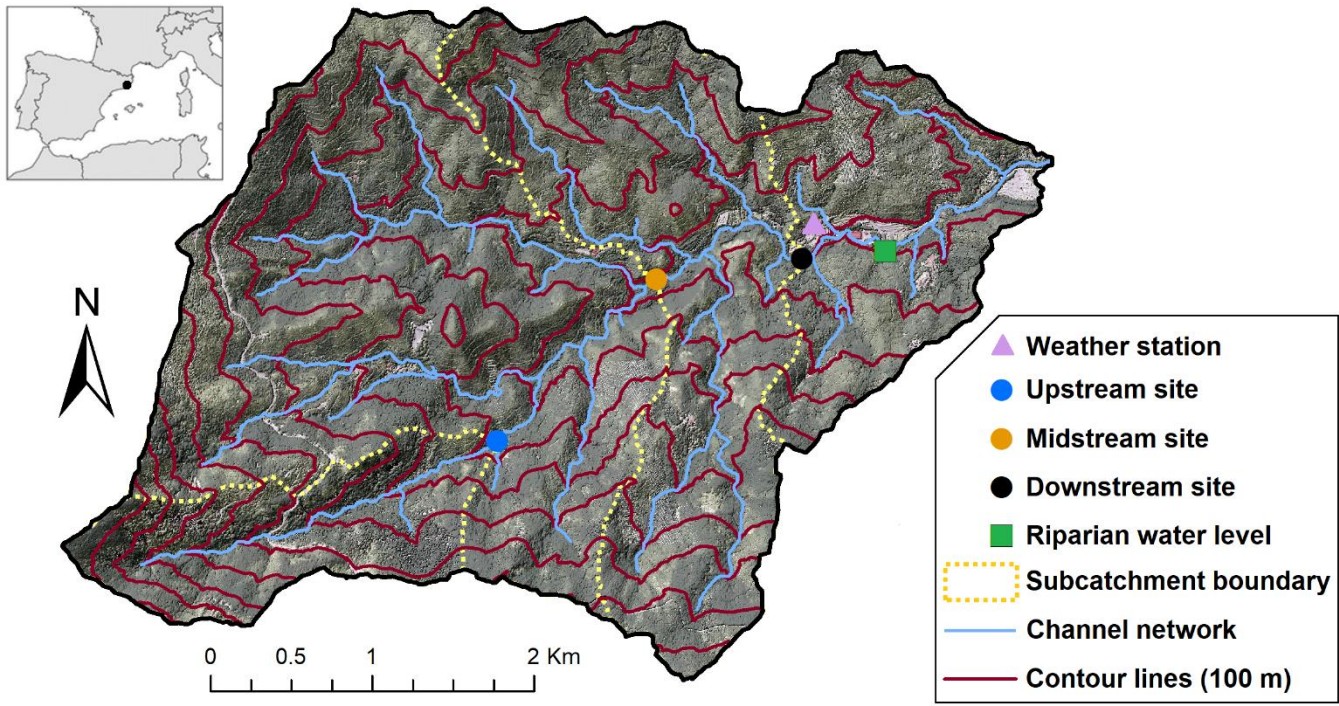

**Figure 1.** Map of the study catchment, Font del Regàs, including weather station, sampling sites, and main hydromorphological properties. The location of the Font del Regàs catchment within Spain is shown in the inset.

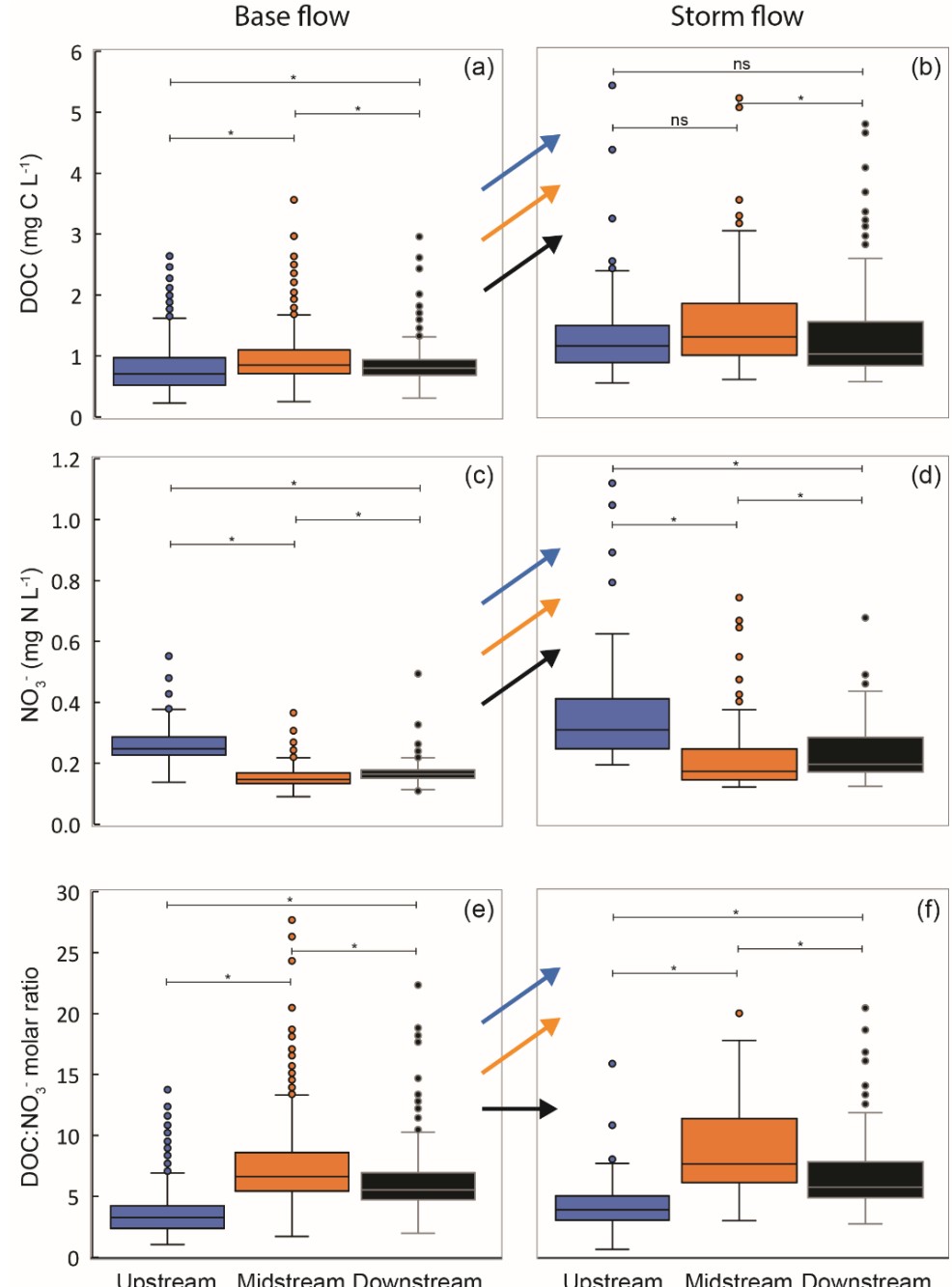

**Figure 2.** Box plots of **(a, b)** dissolved organic carbon (DOC) concentration, **(c, d)** nitrate ($NO_3^-$) concentration, and **(e, f)** DOC:$NO_3^-$ molar ratio for three sampling sites along the Font del Regàs stream for the study period (September 2010 to August 2012). Data are shown separately for base flow (left panels) and storm flow (right panels) conditions. On each box, the central mark indicates the median, and the bottom and top edges of the box indicate the 25th and 75th percentiles, respectively. The whiskers extend to the most extreme data points not considered outliers. Arrows indicate statistical increase (upward arrow) or no statistical difference (straight arrow) between base flow and storm flow conditions for each colour-coded site using non-parametric Wilcoxon rank-sum tests. Results from pairwise Wilcoxon rank-sum tests are also shown, where * indicates statistical difference (p<0.01) and ns indicates non-significant difference.


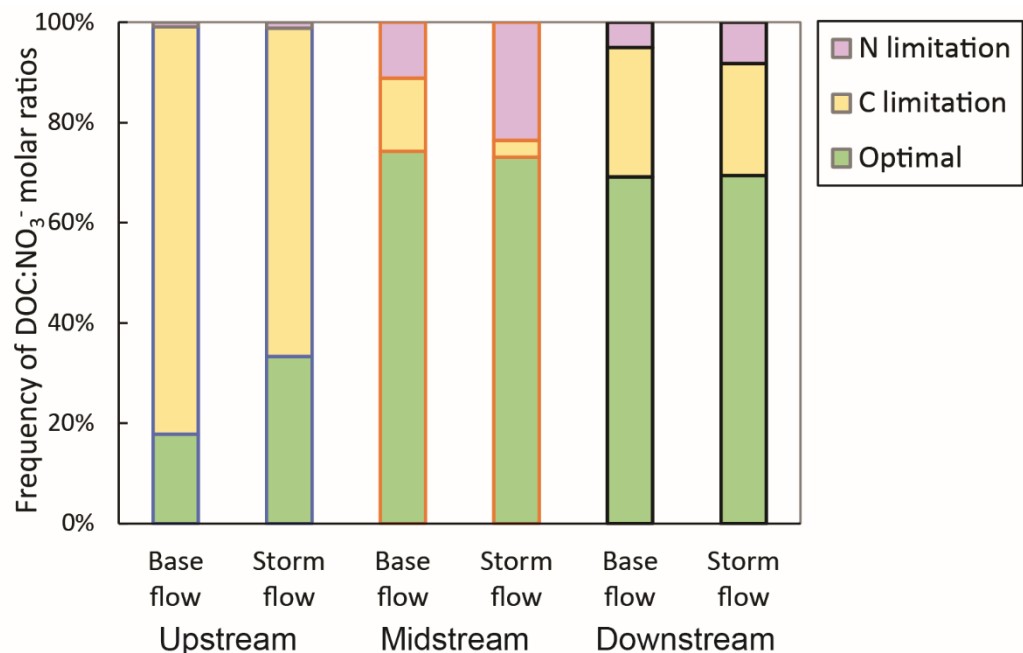

**Figure 3.** Frequency of dissolved organic carbon to nitrate (DOC:NO$_3^-$) molar ratios considered optimal for heterotrophic microorganisms at three sampling sites along the Font del Regàs stream for the study period (September 2010 to August 2012). The corresponding frequencies for which there was either carbon (C) or nitrogen (N) limitation are also displayed in each case. Data are shown separately for base flow and storm flow conditions. The considered range for optimal conditions (4.8 to 11.7) was based on the global reviews of Taylor and Townsend (2010) and Helton et al. (2015).

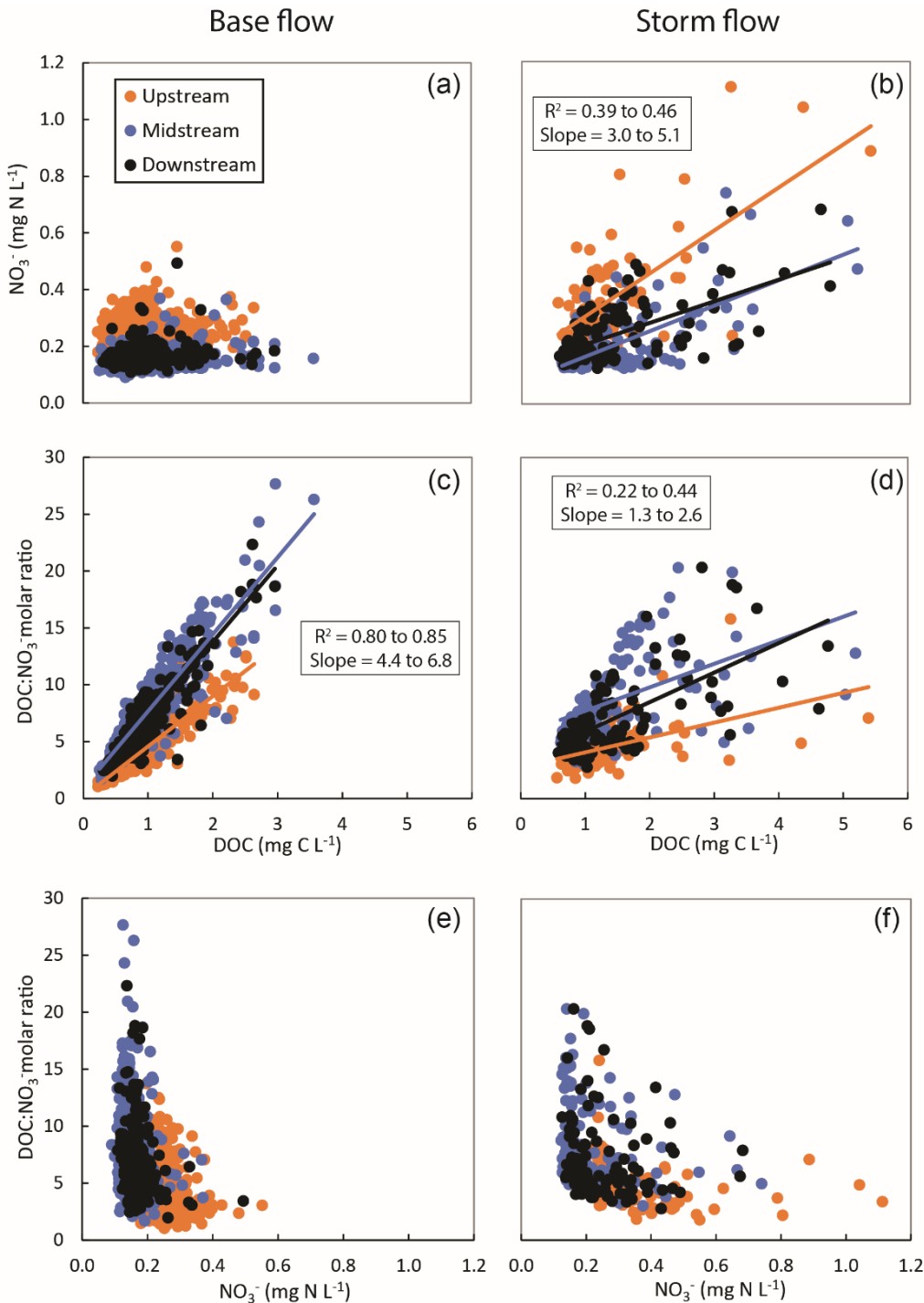


**Figure 4.** Relationships between dissolved organic carbon (DOC) concentration, nitrate ($NO_3^-$) concentration, and DOC:$NO_3^-$ molar ratio for three sampling sites along the Font del Regàs stream for the study period (September 2010 to August 2012). Relationships are shown separately for base flow and storm flow conditions. Coefficients of determination ($R^2$) and slopes are shown for significant linear correlations ($p<0.01$).

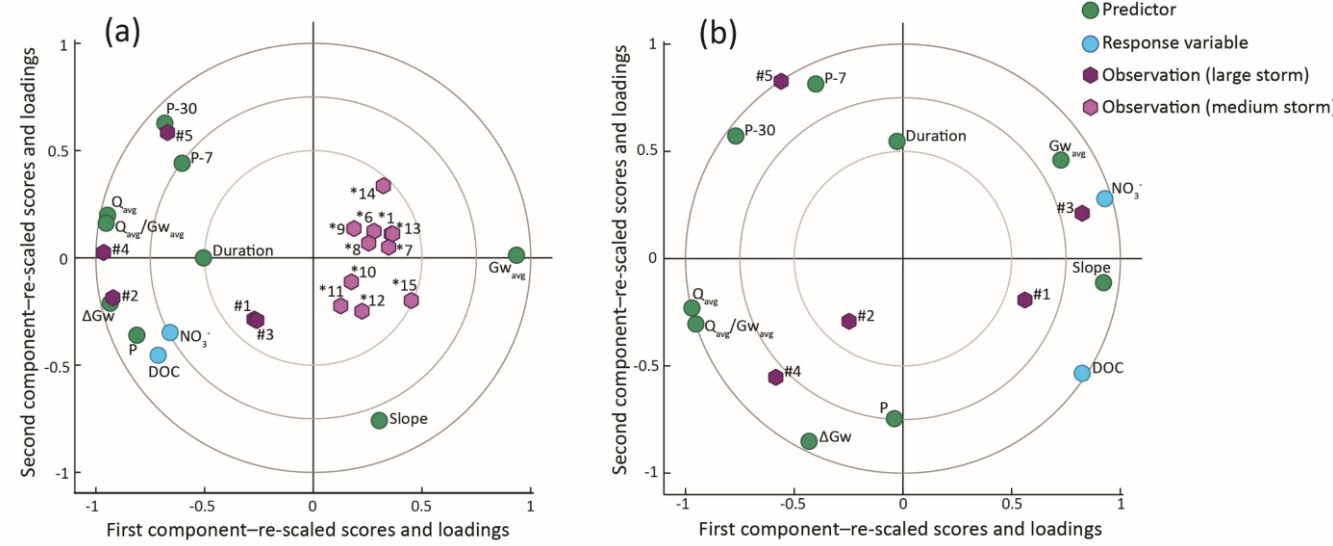

**Figure 5. (a)** Biplot of the two-component partial least square (PLS) regression model with score and loading vectors re-scaled into the -1 to +1 numerical range in order to display the relative relationships between observations (i.e., large and medium storm events), predictors (i.e., duration, accumulated precipitation amount (P), accumulated precipitation seven (P-7) and 30 (P-30) days before the event, average stream flow ($Q_{avg}$), average groundwater table ($Gw_{avg}$), groundwater table range ($\Delta Gw$), slope of the linear relationship between riparian groundwater table and stream flow (slope), and average stream flow normalized to average groundwater table ($Q_{avg}/Gw_{avg}$)), and response variables (average dissolved organic carbon (DOC) concentration and average nitrate ($NO_3^-$) concentration during each event at the downstream site of the Font del Regàs catchment). **(b)** Same biplot for the resulting two-component PLS regression model that only included the large storm events as observations.

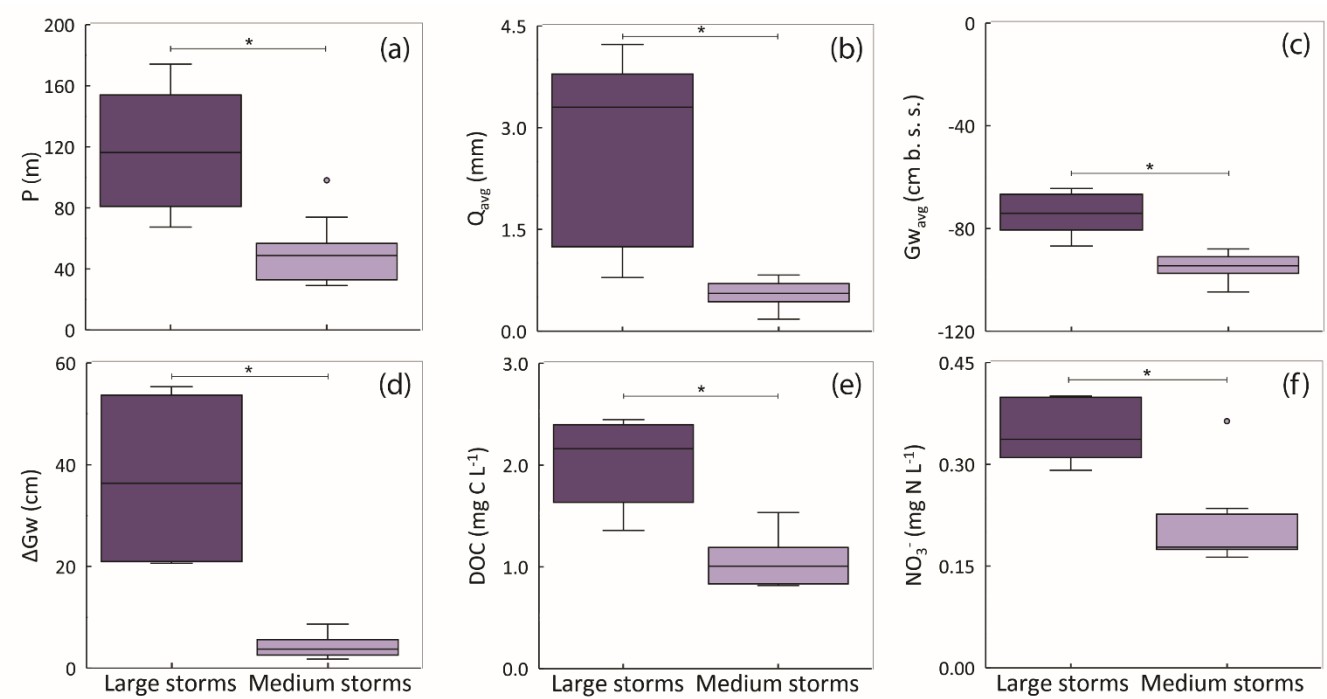

**Figure 6.** Box plots of **(a)** precipitation amount (P), **(b)** average stream flow ($Q_{avg}$), **(c)** average groundwater table ($Gw_{avg}$, with 0 value indicating the soil surface and negative values indicating cm below the soil surface), **(d)** groundwater table range ($\Delta Gw$), **(e)** average dissolved organic carbon (DOC) concentration at the downstream site, and **(f)** average nitrate ($NO_3^-$) concentration at the downstream site for 'large' (n = 5) and 'medium' (n = 11) storm events identified during the study period (September 2010 to August 2012). On each box, the central mark indicates the median, and the bottom and top edges of the box indicate the 25th and 75th percentiles, respectively. The 860 whiskers extend to the most extreme data points not considered outliers. Results from pairwise Wilcoxon rank-sum tests are also shown, where * indicates statistical difference (p<0.01).

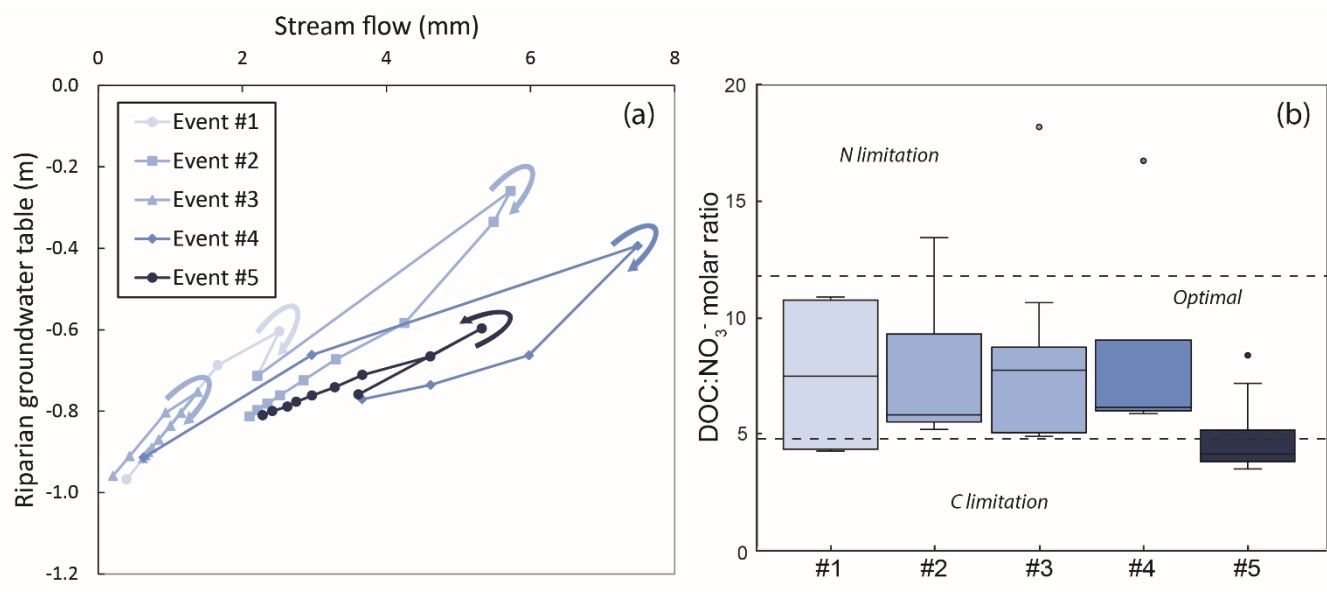

**Figure 7. (a)** Relationships between daily riparian groundwater tables and daily stream flows at the downstream site of the Font del Regàs catchment for the five large storm events identified in the study period (September 2010 to August 2012). All events displayed clockwise hysteresis loops for this relationship (denoted by arrows), except event #5, for which the hysteresis loop was anticlockwise. In all cases, linear regressions showed $R^2>0.8$ (p<0.01). **(b)** Corresponding distribution of dissolved organic carbon to nitrate (DOC:NO$_3^-$) molar ratio in each of the events. On each box, the central mark indicates the median, and the bottom and top edges of the box indicate the 25th and 75th percentiles, respectively. The whiskers extend to the most extreme data points not considered outliers. Dotted lines delimitate the lower (4.8) and upper (11.7) limits of stoichiometric conditions considered optimal for stream heterotrophic microorganisms. In both panels, the darker the blue the wetter the antecedent conditions preceding the corresponding event.



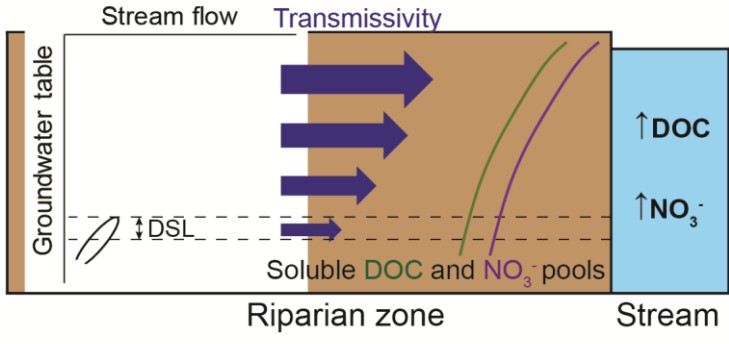

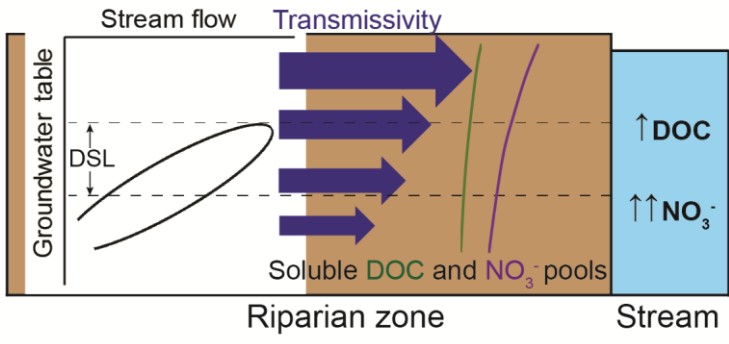

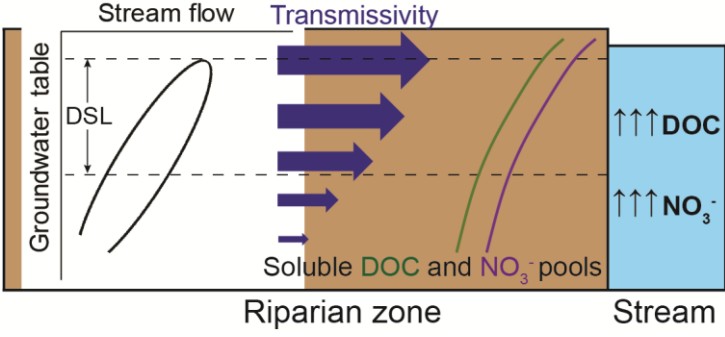

**Figure 8.** Diagram conceptualizing the mobilization of dissolved organic carbon (DOC) and nitrate (NO$_3^-$) from the riparian zone to the stream at the Font del Regàs catchment during **(a)** medium storm events and large storm events preceded by **(b)** wet and **(c)** dry conditions. During large storms preceded by dry antecedent conditions, the riparian groundwater table rises rapidly, as indicated by the steeper slope of the relationship between groundwater table and stream flow. This pattern suggests relatively lower water lateral transmissivity (represented by blue horizontal arrows) of each riparian soil layer compared to large storm events preceded by wet conditions. As a result, the 'Dominant Source Layer', i.e., the riparian zone depth stratum that contributes the most to water and solute fluxes to the stream

(delimitated by dashed horizontal lines), will be thicker during large storms preceded by dry conditions because more layers will be hydrologically activated, increasing the chances for relatively more DOC and $NO_3^-$ mobilization. In addition, soluble DOC and $NO_3^-$ pools (represented by green and purple hypothetical lines, respectively) will accumulate in upper riparian soil layers during dry periods, and will be leached and depleted during very wet periods (especially that of DOC as C is more limited that N in this catchment). As a result, during large storms preceded by dry conditions, groundwater will flow through shallower riparian soil layers with larger DOC and $NO_3^-$ soluble pools, which will lead to large increases in stream DOC and $NO_3^-$ concentrations. During large storms preceded by wet conditions, groundwater will not reach the most superficial soil layers and DOC and $NO_3^-$ pools will be smaller, which will result in minor changes in stream DOC concentration and only moderate increases in stream $NO_3^-$ concentration. During medium storms, antecedent conditions, the slope of the relationship between groundwater table and stream flow, and the size of the soluble DOC and $NO_3^-$ pools in upper riparian layers are of minor relevance because groundwater table elevation is limited, leading to a very narrow DSL and only minor changes in stream DOC and $NO_3^-$ concentrations.

**Appendix A**

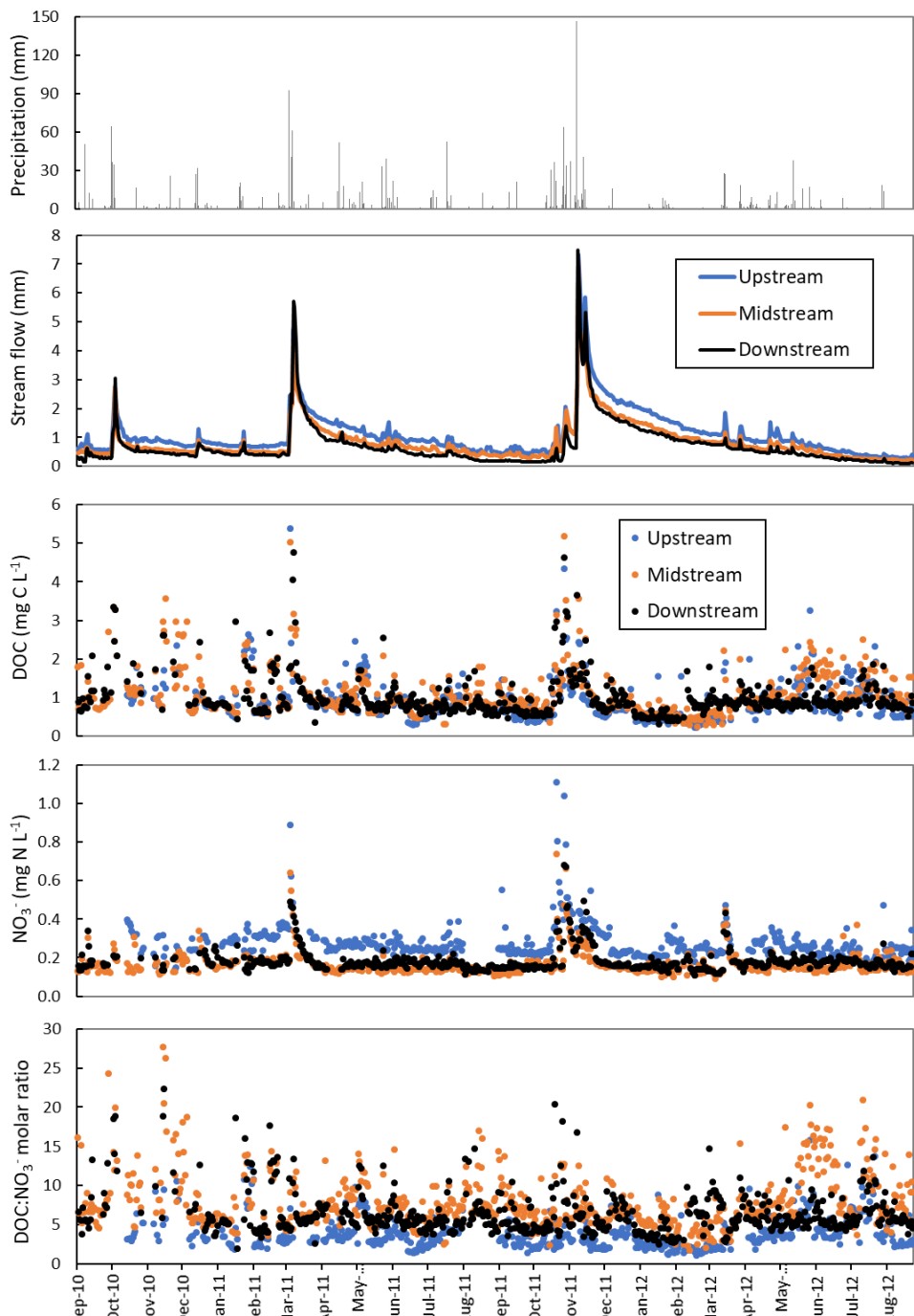


**Figure A1.** Time series of precipitation at Font del Regàs catchment, as well as stream flow, dissolved organic carbon (DOC) concentration, nitrate ($NO_3^-$) concentration, and DOC:$NO_3^-$ molar ratio at three sampling sites (upstream, midstream, and downstream) along the Font Regàs stream for the period September 2010 to August 2012.

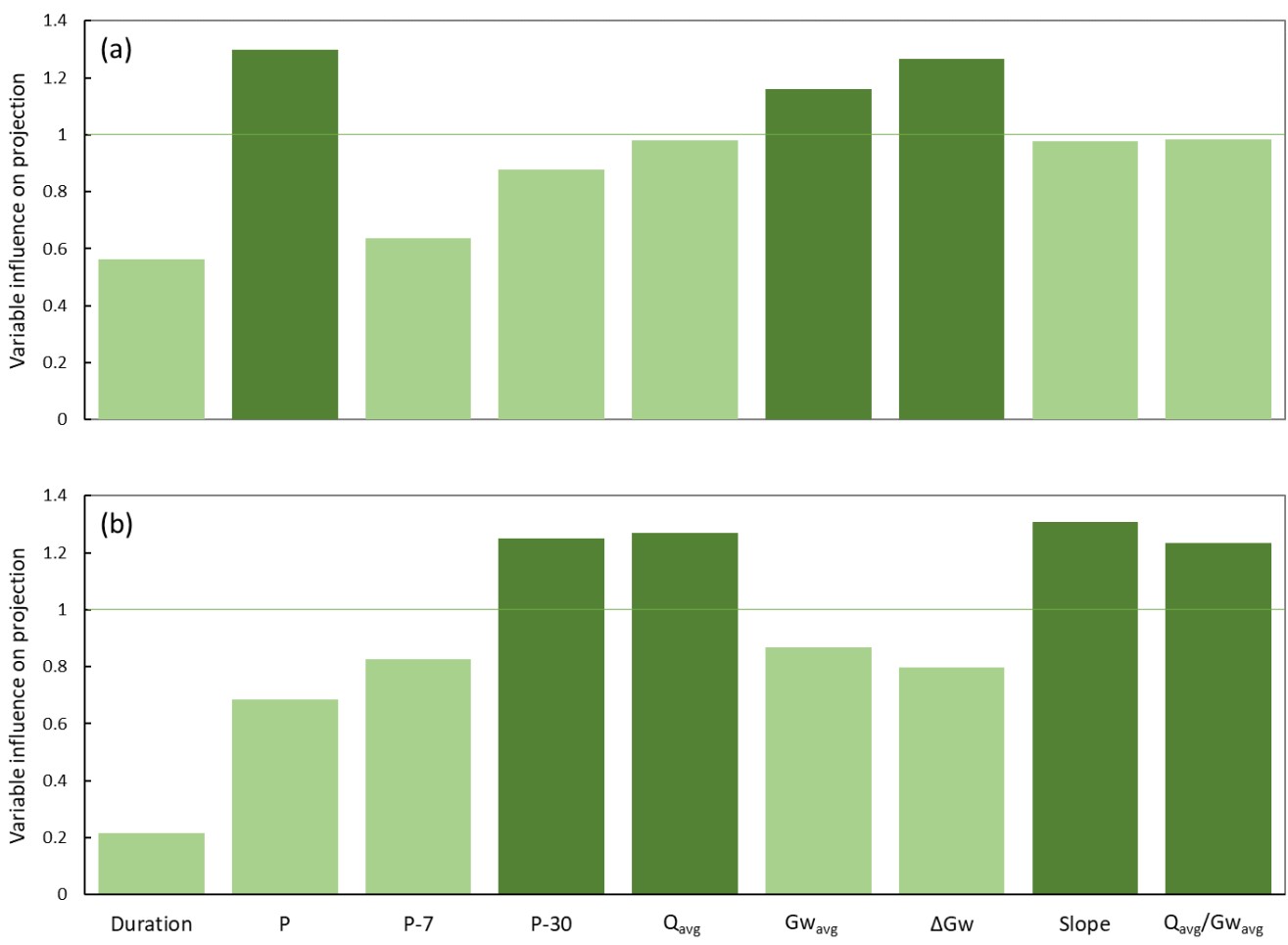


**Figure A2.** VIP (variable influence on projection) values computed for the nine predictors (duration, precipitation amount (P), accumulated precipitation seven (P-7) and 30 (P-30) days before the large storm event, average stream flow ($Q_{avg}$), average groundwater table ($Gw_{avg}$), groundwater table range ($\Delta Gw$), slope of the linear relationship between riparian groundwater table and stream flow (slope), and average stream flow normalized to average groundwater table ($Q_{avg}/Gw_{avg}$)) included in **(a)** the partial least square (PLS) regression

model that included all identified storm events (large and medium) and **(b)** the PLS regression model that only included large storm events. Predictors are considered important in the overall model when VIP>1. Important predictors are highlighted with darker colour.

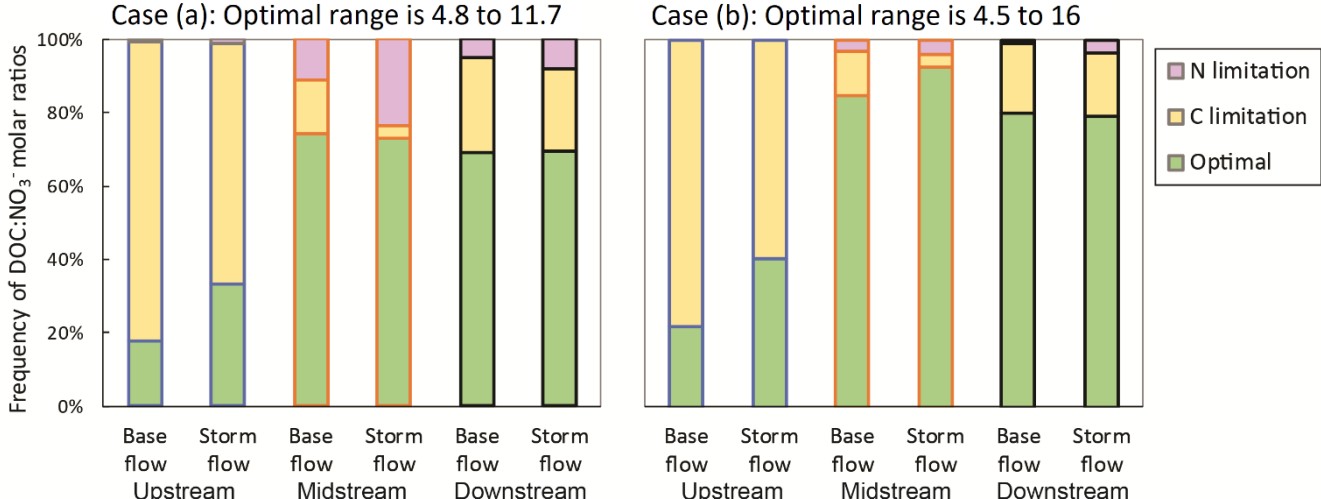

**Figure A3.** Frequency of dissolved organic carbon to nitrate (DOC:NO₃⁻) molar ratios considered optimal for heterotrophic microorganisms at three sampling sites along the Font del Regàs stream for the study period (September 2010 to August 2012) for two considered cases: **(a)** optimal range is 4.8 to 11.7 (presented in the main body of the article and based on the global reviews of Taylor and Townsend (2010) and Helton et al. (2015)) and **(b)** optimal range is 4.5 to 16 (a less restricted range based on values from other modelling and site-specific studies). The corresponding frequencies for which there was either carbon (C) or nitrogen (N) limitation are also displayed. Data are shown separately for base flow and storm flow conditions.
