# Peer review of "Hydrology and riparian forests drive carbon and nitrogen supply and DOC:NO3- stoichiometry along a headwater Mediterranean stream"

_Hydrology and Earth System Sciences, 2021_

## Author Comment (AC1)

**Authors reply to Anonymous Referee #1**

Manuscript title: Hydrology and riparian forests drive carbon and nitrogen supply and DOC:NO$_3^-$ stoichiometry along a headwater Mediterranean stream; by Ledesma, Lupon, Martí, and Bernal

*General comments*

*This study analyses DOC, NO3- and the DOC:NO3- ratio along three stations in a Mediterranean headwater, and across flow conditions, during a 2-year period. Spatial variability is controlled by the presence of riparian forest and topography, while temporal variability is controlled by hydroclimatic conditions. The authors conclude that this spatiotemporal variability influences stream metabolic processes.*

*The manuscript is well written and the conclusions are clear. This work is within the scope of HESS but I find this is a modest contribution to the literature.*

*Here are three options to make a stronger paper:*

- *Relax the selection criteria for the storm events to include in the PLS regression (currently only 5 observations)*
- *Include data on DOC and N composition, not only in the discussion. The discussion suggests that such data is available. Is nitrate the only N form in this stream?*
- *Include more recent data to make a more complete synthesis of research in this catchment. The references indicate that other studies have taken place in this catchment since the monitoring period 2010-2012 considered here.*

**[Reply]**: We are happy that you found the manuscript "well written" and the conclusions "clear" and thank you for the three suggestions to make the paper stronger. They are sensible and, in fact, we had already considered these options during the preparation of the manuscript. Below we disclose relevant information and data related to each of these points and argue about the actions that we will take in each case.

Including more storm events in the PLS regression

Following your suggestion, we have relaxed the selection criteria for including storm events in our analyses. For this exercise, new events were included if they fulfilled two requirements: (i) a precipitation amount during the days included in the event of at least 25 mm (events below these threshold showed no groundwater table response at the temporal resolution of the study and marginal stream flow responses), and (ii) stream flow was classified as storm flow during the days of the event (not all events with precipitation amounts higher than 25 mm generated storm flow at the temporal resolution of the study).

Using these more relaxed criteria, a total of 15 new events were identified. The same nine hydroclimatic descriptors used in Table 2 of the manuscript for the former events, together with the average DOC and

NO$_3^-$ concentrations during each event and the relative increase in DOC and NO$_3^-$ concentrations with respect to base flow conditions (within brackets), are shown in Table R1 below for these new events.

*Table R1. Hydroclimatic descriptors of the new 11 medium storm events identified during the study period*

| Event | Period | Duration (days) | P (mm) | P-7 (mm) | P-30 (mm) | Q$_{avg}$ (mm) | Gw$_{avg}$ (m) | ΔGw (m) | Slope (m mm$^{-1}$) | Q$_{avg}$/Gw$_{avg}$ (mm m$^{-1}$) | DOC (mg C L$^{-1}$) | NO$_3^-$ (mg N L$^{-1}$) |
|---|---|---|---|---|---|---|---|---|---|---|---|---|
| *1 | 16-19/09/2010 | 4 | 52 | 9 | 36 | 0.4 | 0.97 | 0.04 | 0.10 | 0.4 | 1.01 (15%) | 0.23 (36%) |
| *2 | 09-17/10/2010 | 9 | 147 | 6 | 88 | 1.3 | 0.87 | 0.27 | 0.10 | 1.5 | na | na |
| *3 | 21-29/12/2010 | 9 | 67 | 1 | 44 | 0.6 | 0.93 | 0.09 | 0.19 | 0.7 | na | na |
| *4 | 27/01-02/02/2011 | 7 | 57 | 0 | 20 | 0.5 | 0.96 | 0.03 | 0.05 | 0.6 | na | na |
| *5 | 24-27/04/2011 | 4 | 53 | 15 | 44 | 1.0 | 0.88 | 0.02 | -0.02 | 1.1 | na | na |
| *6 | 14-17/05/2011 | 4 | 30 | 22 | 122 | 0.7 | 0.90 | 0.02 | 0.15 | 0.8 | 0.99 (13%) | 0.17 (5%) |
| *7 | 30/05-02/06/2011 | 4 | 36 | 0 | 69 | 0.6 | 0.92 | 0.02 | 0.15 | 0.6 | 1.19 (36%) | 0.18 (7%) |
| *8 | 03-06/06/2011 | 4 | 50 | 36 | 96 | 0.6 | 0.92 | 0.03 | 0.14 | 0.7 | 0.83 (-5%) | 0.17 (-1%) |
| *9 | 09-15/06/2011 | 7 | 34 | 66 | 148 | 0.6 | 0.91 | 0.03 | 0.15 | 0.7 | 1.1 (26%) | 0.18 (7%) |
| *10 | 25/07-03/08/2011 | 10 | 74 | 4 | 48 | 0.5 | 0.94 | 0.04 | 0.16 | 0.5 | 0.82 (-7%) | 0.18 (5%) |
| *11 | 23-30/10/2011 | 8 | 98 | 17 | 39 | 0.3 | 0.95 | 0.09 | 0.17 | 0.3 | 1.53 (75%) | 0.23 (41%) |
| *12 | 21-23/03/2012 | 3 | 57 | 6 | 8 | 0.8 | 0.88 | 0.06 | 0.22 | 0.9 | 0.95 (8%) | 0.36 (118%) |
| *13 | 03-05/04/2012 | 3 | 29 | 0 | 63 | 0.7 | 0.95 | 0.06 | 0.13 | 0.7 | 1.23 (40%) | 0.16 (-2%) |
| *14 | 18-24/05/2012 | 7 | 49 | 10 | 62 | 0.5 | 0.97 | 0.04 | 0.02 | 0.5 | 0.82 (-7%) | 0.19 (12%) |
| *15 | 04-09/08/2012 | 6 | 33 | 0 | 4 | 0.2 | 1.05 | 0.03 | 0.23 | 0.2 | 1.08 (23%) | 0.18 (5%) |

na: not available or incomplete

Unfortunately, four of the events (*2, *3, *4, and *5) had no available chemical data associated with them or these data were incomplete. The ultimate goal of our PLS regression analysis was to relate hydroclimatic characteristics of storm events with resulting DOC and NO$_3^-$ concentrations in the stream and thus these four events cannot be integrated into this analysis because they lack relevant information. This is the reason why we introduced the requirement "a complete stream chemistry data series associated with the event for the downstream site (i.e. no gaps in chemical data for the event dates)" (L. 185-186) for including events in the analysis in the original manuscript. These four events will not be further discussed.

From the remaining new 11 events, hereafter referred as "medium storm events", (i) 82% (9 out of 11) accumulated lower precipitation amounts than the lowest of the precipitation amounts of the former large storm events (Figure R1a below), (ii) 91% (10 out of 11) sowed lower average stream flow than the lowest of the average stream flows of the former large storm events (Figure R1b), (iii) 100% displayed deeper groundwater tables than the deepest of all groundwater tables of the former large storm events (Figure R1c), and (iv) 100% exhibited smaller groundwater table ranges (i.e. the thickness of the riparian layer that becomes hydrologically activated during the event) than the smallest of the groundwater table ranges of the former large storm events (Figure R1d).

[Figure]

*Figure R1. Histograms of the hydroclimatic descriptors (a) precipitation amount (P), (b) average stream flow (Q$_{avg}$), (c) average groundwater table (Gw$_{avg}$), and (d) groundwater table range (ΔGw) for the 5 large storm events included in the former version of the manuscript ("large storms") and the 11 new medium-size events ("medium storms").*

Importantly, these differences in climatic and, especially, hydrological chatacteristics between the large and medium storm events led to marked differences in the resulting stream chemistry: both DOC and NO$_3^-$ concentrations were substantially lower during the medium storm events than during the large storm events (Figure R2 below). This observed pattern is likely caused by the hydrological activation of a thicker and shallower riparian layer during large storm events that leads to the mobilization of relatively larger amounts of DOC and NO$_3^-$ stored in the riparian soil compared to the amounts mobilized from the deeper and narrower layers that are hydrologically activated during medium storm events. These results suggest that large and medium storm events display a distinct mechanism of DOC and NO$_3^-$ mobilization from the riparian zone that precludes a direct integration of the samples from the two type of events into the PLS regression analysis and into the conceptual model we propose in Figure 7 of the manuscript.

[Figure]

[Figure]

*Figure R2. Box plots of dissolved organic carbon (DOC) concentrations and nitrate ($NO_3^-$) concentrations in stream waer for the 5 large storm events included in the former version of the manuscript ("large storms") and the 11 new medium-size events ("medium storms").*

[Figure]

*Figure R3. Biplot of a two-component partial least square (PLS) regression model with score and loading vectors re-scaled into the -1 to +1 numerical range in order to display the relative relationships between observations (i.e. 5 large storm events versus 11 medium storm events), predictors, and response variables.*

To illustrate our point further, we have also reanalyzed the PLS regression model including both the former large (N = 5, denoted with #) and the new medium (N = 11, denoted with *) storm events (Figure R3 above). The relative ordination of the predictors is similar to the original PLS regression model, with *P-30*, $Q_{avg}$, and $Q_{avg}/Gw_{avg}$ located relatively contigous in one side of the ordination and *slope* located in the opposite side. However, compared to the original analysis, the model goodness of fit is reduced ($R^2Y$ decreased from 0.96 to 0.72 for DOC and from 0.94 to 0.55 for $NO_3^-$) and the model predictive ability is compromised ($Q^2Y$ decreased from 0.88 to 0.49 for DOC and from 0.82 to 0.25 for $NO_3^-$). Remarkably, the two type of storm events fall in two opposite regions of the biplot: medium storm events cluster in the right side of the ordination, while large storm events make a broader cluster in the left side of the ordination. This result further demonstrate the different nature of the two type of events and the different implications for DOC and $NO_3^-$ mobilization.

All in all, in the revised manuscript we will keep the five large storm events in our PLS regression model, which is the basis for our suggested conceptual model that only applies to such large events. Nevertheless, we will also underscore the hydrological differences between large and medium storm events and discuss the biogeochemical implications of such differences in terms of DOC and $NO_3^-$ mobilization and concentrations in the stream. For that, we will integrate in the main body and the supplementary material of the revised manuscript a condensed version of the information we have presented in this document in this regard.

DOM composition and other forms of nitrogen

Unfortunately, we do not have data on DOM composition at daily resolution or during storm flow conditions, which prevents us for directly integrating DOM composition data into the analyses of the present study. Instead, during the same study period (2010-2012), we performed longitudinal surveys of DOM composition on 11 occasions only during base flow conditions. The data from these surveys showed that DOM in Font del Regàs has a prominent protein-like character in both riparian groundwater and stream water and we published these results in a previous study (Bernal et al. 2018). Here we use this information by referring to the published study to support our suggestion that in-stream heterotrophic activity could be partially sustained during storm flow conditions. We will reword the sentence in the discussion where we included this information to make clear that the data we have on DOM composition is from base flow conditions and that we assume the character is maintained across flow conditions as "Further, another study from Font del Regàs showed that DOM has a prominent protein-like character in both riparian groundwater and stream water during base flow conditions (Bernal et al., 2018), which could lead to rapid assimilation even during periods of short water residence times associated with storm flow conditions, assuming DOM molecular composition is maintained across flow conditions".

Regarding the different forms of nitrogen, $NO_3^-$ is not the only form found at Font del Regàs but it makes up the overwhelming majority of both inorganic and total nitrogen. For the study period (2010-2012), we do have data on daily concentrations of other forms of dissolved inorganic nitrogen (DIN), including $NH_4^+$ and $NO_2^-$ (partially published in Lupon et al., 2016b). Overall average stream concentrations of $NH_4^+$

(0.01 ± 0.006 mg N $L^{-1}$) and $NO_2^-$ (0.006 ± 0.005 mg N $L^{-1}$) are significantly lower than overall average stream $NO_3^-$ concentrations (0.20 ± 0.09 mg N $L^{-1}$). Moreover, $NH_4^+$ and $NO_2^-$ concentrations are in all cases lower than 0.02 mg N $L^{-1}$ and show no differences between baseflow and storm flow conditions. Hence, $NO_3^-$ accounted for more than 90% of DIN during both base flow and storm flow during the study period. It its turn, dissolved organic nitrogen (DON) was always below 0.05 mg mg N $L^{-1}$, which implies that $NO_3^-$ makes up more than 80% of the total dissolved nitrogen under all circumstances. Given that (i) $NO_3^-$ is the major source of nitrogen for stream heterotrophic microorganisms in our system, and (ii) no significant differences were observed in other DIN forms between base and storm flow conditions, we decided to only analyze spatiotemporal patterns in $NO_3^-$ concentrations. We will add this information in the revised manuscript.

Including more recent data

Unfortunately, we do not have chemistry data at the temporal (i.e. daily) or spatial (i.e. three sites along the stream) resolutions that we used in the present study beyond the period that we have analysed here. Some of the studies from the catchment that we cite and that you refer to did in fact use parts of the data that we present here in order to answer different questions, and thus do not include data beyond 2012. Others use data from specific experiments or campaigns that cannot be directly incorporated into our analyses, e.g. the longitudinal surveys of DOM composition presented in Bernal et al. (2018) that we discuss above. In autumn 2018, we started a mostly hydrological monitoring effort at the downstream site and DOC and $NO_3^-$ concentrations (and consequently DOC:$NO_3^-$ molar ratios), which are the main focus of the present study, have only been measured sporadically in the stream water in the downstream site or during focused short experiments along the streams. Therefore, there is no other data outside the study period that can be integrated into our analyses and we only use other studies on Font del Regàs to support our discussion.

*Specific comments*

*Figure 1: add location of the weather station*

**[Reply]**: Thanks for pointing this out. We will add the location of the automatic weather station in a revised Figure 1.

*L115 "Rating curves obtained from the relationships between stream flow and stream water level measurements were used to construct daily time series of stream flow data at each site" can you provide the rating curves in SI?*

**[Reply]**: The rating curves were presented in the supplementary material of Lupon et al. (2016b), published also in *HESS*, and we can provide them as a new figure in the Appendix of the present study if the Editor thinks this is adequate.

*L122 "the dynamics of this dataset capture well the dynamics of the groundwater table variation in the surrounding riparian area and therefore we are confident that the recorded pattern at the monitoring*

*location was representative of the groundwater table variations in the riparian zone" please provide stronger evidence that this piezometer is representative of the whole downstream area.*

**[Reply]**: The evidence for this statement was presented in Ledesma et al. (2021). In addition, to the groundwater tables presented in this study (recorded at 15 min intervals using a water pressure transducer installed in a piezometer placed 2.5 m away from the stream channel), we also measured groundwater tables manually every two weeks during the same period at seven equidistant (ca. 3 m) piezometers placed ca. 2 m from the stream channel along the same area where the pressure transducer was located. We compared a total of 45 supplementary groundwater table measurements available at each of the seven piezometers with the data from the pressure transducer, and showed that groundwater table dynamics were notably similar in all cases, which supports the use of the pressure transducer data as representative of the downstream riparian areas in our study. The figure below shows the comparison of pressure transducer versus manually measured groundwater tables (measurements from each piezometer are presented in a different colour and dotted lines are linear regressions between the two variables for each corresponding colour code case (p<0.0001 in all cases)):

[Figure]

In the revised manuscript, we will rephrase the sentence to make the evidence more explicit as "In a previous study, we showed that the dynamics of the pressure transducer dataset captured well the dynamics of manually-measured groundwater table variations in seven other piezometers located in the surrounding riparian area (Ledesma et al., 2021). Therefore, we are confident that the recorded pattern at the pressure transducer location was representative of the groundwater table variations in the riparian zone soils in the lower parts of the catchment".

*L174 "hydroclimatic analysis of large storm events" I understand that the authors chose to analyze the largest storm events because they probably exhibit the clearest signal, but the selection criteria here are very strict and only 5 storm events were kept for analysis. This is a very small number, even though PLS regression can handle datasets with few observations and many variables. Wouldn't it be more interesting to relax the selection criteria and include more storm events?*

**[Reply]**: Please, see our detailed response to this issue above.

*L322 "given that this is a predominantly heterotrophic system (Lupon et al., 2016c)." please explain how this was determined (most readers won't read the reference)*

**[Reply]**: The stream is predominantly heterotrophic because daily rates of ecosystem respiration (5.0 – 10.0 g $O_2$ $m^{-2}$ $day^{-1}$) are between 10- and 100-fold higher than daily rates of gross primary production (0.1 – 0.7 g $O_2$ $m^{-2}$ $day^{-1}$), as we measured and reported in Lupon et al. (2016c). Following your suggestion, we will include this information in the revised manuscript as "[…] given that this is a predominantly heterotrophic system, i.e. rates of ecosystem respiration are between 10- and 100-fold higher than rates of gross primary production (Lupon et al., 2016c)".

*L340 "This result is in line with the idea that headwater streams can remove substantial amounts of NO3 - within relatively short distances (Peterson et al., 2001) […] providing groundwater inputs with low NO3 - concentrations driven by denitrification, as observed in temperate forest catchments (Cirmo and McDonnell, 1997)." Both instream removal and dilution from the middle part of the catchment can explain this decrease. Is it possible to estimate the share of each process?*

**[Reply]**: In general, the data presented in this study cannot be analyzed and evaluated in a way that would allow estimating the relative contribution of *in-stream removal* vs. *dilution from riparian denitrification* to the $NO_3^-$ decrease between the upper and midstream subcatchments. Nevertheless, in this case we are confident than *in-stream removal* overwhelmingly dominates over *dilution from riparian denitrification* because the riparian soils at Font del Regàs do not support high denitrification rates (< 3 µg N $kg^{-1}$ $day^{-1}$; Poblador et al., 2017), but rather sustain large net nitrification rates (1 – 2 mg N $kg^{-1}$ $day^{-1}$; Lupon et al., 2016a). Yet, even if *in-stream removal* likely accounts for most of the $NO_3^-$ decrease, we consider that this process alone cannot explain the entire $NO_3^-$ reduction and we propose that "the steep topography of the upstream subcatchment led to rapid drainage of aerated [hillslope] soils and relatively larger $NO_3^-$ mobilization compared to the flatter near-stream zones of the midstream subcatchment" (L. 343-345). Therefore, the second mechanism playing a role on the $NO_3^-$ decrease would be a comparatively lower $NO_3^-$ input in the middle part of the catchment relative to the upper part of the catchment driven by topography, rather than a dilution driven by riparian denitrification.

*L375 "The magnitude of change between flow conditions was different for DOC and NO3 - at the upstream site…" please specify which of DOC or NO3- increases more.*

**[Reply]**: We will specify this information in the revised manuscript as "The magnitude of change between flow conditions was different for DOC (which on average increased by 66%) and $NO_3^-$ (which on average increased by 41%) at the upstream site, leading to an increase in the frequency of optimal DOC:$NO_3^-$ stoichiometric conditions for heterotrophic activity during storm flow".

*L395 "another study from Font del Regàs showed that DOM has a prominent protein-like character in both riparian groundwater and stream water (Bernal et al., 2018)" suggest to include this data in the analysis (not only just in the discussion) to make a more complete paper. The speciation of DOC and the N species other than nitrate should be analyzed further.*

**[Reply]**: Please, see our detailed responses to these issues above.

*Technical corrections*

*L110 "All data and analyses were integrated and carried out for daily resolutions, which were determined by the availability of the stream chemical data." This sentence is unclear*

**[Reply]**: We agree, and in the revised manuscript we will change this sentence to "All data and analyses were integrated and carried out for daily resolutions, which was the resolution of the stream chemical data".

**References used in this authors reply**

Bernal, S., Lupon, A., Catalán, N., Castelar, S., and Martí, E.: Decoupling of dissolved organic matter patterns between stream and riparian groundwater in a headwater forested catchment, Hydrol. Earth Syst. Sci., 22, 1897-1910, https://doi.org/10.5194/hess-22-1897-2018, 2018.

Ledesma, J. L. J., Ruiz-Pérez, G., Lupon, A., Poblador, S., Futter, M. N., Sabater, F., and Bernal, S.: Future changes in the Dominant Source Layer of riparian lateral water fluxes in a subhumid Mediterranean catchment. J. Hydrol., 595, 126014, https://doi.org/10.1016/j.jhydrol.2021.126014, 2021.

Lupon, A., Sabater, F., Miñarro, A., and Bernal, S.: Contribution of pulses of soil nitrogen mineralization and nitrification to soil nitrogen availability in three Mediterranean forests, Eur. J. Soil Sci., 67, 303-313, https://doi.org/10.1111/ejss.12344, 2016a.

Lupon, A., Bernal, S., Poblador, S., Martí, E., and Sabater, F.: The influence of riparian evapotranspiration on stream hydrology and nitrogen retention in a subhumid Mediterranean catchment, Hydrol. Earth Syst. Sci., 20, 3831-3842, https://doi.org/10.5194/hess-20-3831-2016, 2016b.

Lupon, A., Martí, E., Sabater, F., and Bernal, S.: Green light: gross primary production influences seasonal stream N export by controlling fine-scale N dynamics, Ecology, 97, 133-144, https://doi.org/10.1890/14-2296.1, 2016c.

Poblador, S., Lupon, A., Sabaté, S., and Sabater, F.: Soil water content drives spatiotemporal patterns of $CO_2$ and $N_2O$ emissions from a Mediterranean riparian forest soil, Biogeosciences, 14, 4195-4208, 10.5194/bg-14-4195-2017, 2017.

---

## Author Comment (AC2)

**Authors reply to Anonymous Referee #2**

Manuscript title: Hydrology and riparian forests drive carbon and nitrogen supply and DOC:NO$_3^-$ stoichiometry along a headwater Mediterranean stream; by Ledesma, Lupon, Martí, and Bernal

*General comments*

*This manuscript explores hydrological and riparian controls on DOC and NO3- stream chemistry in an oligotrophic Mediterranean stream. Their rationale was to understand how these dynamics may impact in-stream heterotrophic activity. To do this, the authors use data collected at three sites within the catchment collected over a period 2 years (2010-2012). Their results suggest that spatial (upstream to downstream) patterns are explained by riparian geomorphology and forest coverage and that temporal variability results from hydrological variability. The manuscript is well written, figures are clear, and conclusions are straightforward and supported by data. I particularly enjoyed the use of the hydrological metrics to understand controls on lateral exchanges of DOC and NO3- between the riparian zone. That said, the contributions to the literature are modest. I wonder if the authors could strengthen the manuscript by adding data beyond 2010-2012 and include more (smaller) storm events in their PLSR analysis. Large storms are relatively rare and DOC and NO3- transport during smaller, and likely more frequent, storms may influence stream metabolism more than the larger events. Also, I suspect antecedent conditions are even more important for DOC and NO3- transport dynamics during these smaller events. I also wonder if the authors have NH4+ and/or DON concentrations and might consider looking at the DOC:DIN or DOC:TDN ratios. Perhaps NO3- is the dominant form of N and these other N forms won't impact the story much?*

**[Reply]**: We are happy that you found the manuscript "well written" and the conclusions "straightforward and supported by data" and we acknowledge the appreciation of the use of hydrological metrics. Please, see below our detail responses to the three main points we identified in this general comment.

Including more storm events in the PLS regression

Following your suggestion, we have explored the role of not only the large storm events, but also of smaller events on DOC and NO$_3^-$ mobilization in the context of the present study. For that, we relaxed the selection criteria for including storm events in our analyses. For this exercise, new events were included if they fulfilled two requirements: (i) a precipitation amount during the days included in the event of at least 25 mm (events below these threshold showed no groundwater table response at the temporal resolution of the study and marginal stream flow responses), and (ii) stream flow was classified as storm flow during the days of the event (not all events with precipitation amounts higher than 25 mm generated storm flow at the temporal resolution of the study).

Using these more relaxed criteria, a total of 15 new events were identified. The same nine hydroclimatic descriptors used in Table 2 of the manuscript for the former events, together with the average DOC and NO$_3^-$ concentrations during each event and the relative increase in DOC and NO$_3^-$ concentrations with respect to base flow conditions (within brackets), are shown in Table R1 below for these new events.

*Table R1. Hydroclimatic descriptors of the new 11 medium storm events identified during the study period*

| Event | Period | Duration (days) | P (mm) | P-7 (mm) | P-30 (mm) | $Q_{avg}$ (mm) | $Gw_{avg}$ (m) | $\Delta Gw$ (m) | Slope (m mm$^{-1}$) | $Q_{avg}/Gw_{avg}$ (mm m$^{-1}$) | DOC (mg C L$^{-1}$) | NO$_3^-$ (mg N L$^{-1}$) |
|---|---|---|---|---|---|---|---|---|---|---|---|---|
| *1 | 16-19/09/2010 | 4 | 52 | 9 | 36 | 0.4 | 0.97 | 0.04 | 0.10 | 0.4 | 1.01 (15%) | 0.23 (36%) |
| *2 | 09-17/10/2010 | 9 | 147 | 6 | 88 | 1.3 | 0.87 | 0.27 | 0.10 | 1.5 | na | na |
| *3 | 21-29/12/2010 | 9 | 67 | 1 | 44 | 0.6 | 0.93 | 0.09 | 0.19 | 0.7 | na | na |
| *4 | 27/01-02/02/2011 | 7 | 57 | 0 | 20 | 0.5 | 0.96 | 0.03 | 0.05 | 0.6 | na | na |
| *5 | 24-27/04/2011 | 4 | 53 | 15 | 44 | 1.0 | 0.88 | 0.02 | -0.02 | 1.1 | na | na |
| *6 | 14-17/05/2011 | 4 | 30 | 22 | 122 | 0.7 | 0.90 | 0.02 | 0.15 | 0.8 | 0.99 (13%) | 0.17 (5%) |
| *7 | 30/05-02/06/2011 | 4 | 36 | 0 | 69 | 0.6 | 0.92 | 0.02 | 0.15 | 0.6 | 1.19 (36%) | 0.18 (7%) |
| *8 | 03-06/06/2011 | 4 | 50 | 36 | 96 | 0.6 | 0.92 | 0.03 | 0.14 | 0.7 | 0.83 (-5%) | 0.17 (-1%) |
| *9 | 09-15/06/2011 | 7 | 34 | 66 | 148 | 0.6 | 0.91 | 0.03 | 0.15 | 0.7 | 1.1 (26%) | 0.18 (7%) |
| *10 | 25/07-03/08/2011 | 10 | 74 | 4 | 48 | 0.5 | 0.94 | 0.04 | 0.16 | 0.5 | 0.82 (-7%) | 0.18 (5%) |
| *11 | 23-30/10/2011 | 8 | 98 | 17 | 39 | 0.3 | 0.95 | 0.09 | 0.17 | 0.3 | 1.53 (75%) | 0.23 (41%) |
| *12 | 21-23/03/2012 | 3 | 57 | 6 | 8 | 0.8 | 0.88 | 0.06 | 0.22 | 0.9 | 0.95 (8%) | 0.36 (118%) |
| *13 | 03-05/04/2012 | 3 | 29 | 0 | 63 | 0.7 | 0.95 | 0.06 | 0.13 | 0.7 | 1.23 (40%) | 0.16 (-2%) |
| *14 | 18-24/05/2012 | 7 | 49 | 10 | 62 | 0.5 | 0.97 | 0.04 | 0.02 | 0.5 | 0.82 (-7%) | 0.19 (12%) |
| *15 | 04-09/08/2012 | 6 | 33 | 0 | 4 | 0.2 | 1.05 | 0.03 | 0.23 | 0.2 | 1.08 (23%) | 0.18 (5%) |

na: not available or incomplete

Unfortunately, four of the events (*2, *3, *4, and *5) had no available chemical data associated with them or these data were incomplete. The ultimate goal of our PLS regression analysis was to relate hydroclimatic characteristics of storm events with resulting DOC and NO$_3^-$ concentrations in the stream and thus these four events cannot be integrated into this analysis because they lack relevant information. This is the reason why we introduced the requirement "a complete stream chemistry data series associated with the event for the downstream site (i.e. no gaps in chemical data for the event dates)" (L. 185-186) for including events in the analysis in the original manuscript. These four events will not be further discussed.

From the remaining new 11 events, hereafter referred as "medium storm events", (i) 82% (9 out of 11) accumulated lower precipitation amounts than the lowest of the precipitation amounts of the former large storm events (Figure R1a below), (ii) 91% (10 out of 11) sowed lower average stream flow than the lowest of the average stream flows of the former large storm events (Figure R1b), (iii) 100% displayed deeper groundwater tables than the deepest of all groundwater tables of the former large storm events (Figure R1c), and (iv) 100% exhibited smaller groundwater table ranges (i.e. the thickness of the riparian layer that becomes hydrologically activated during the event) than the smallest of the groundwater table ranges of the former large storm events (Figure R1d).

[Figure]

*Figure R1. Histograms of the hydroclimatic descriptors (a) precipitation amount (P), (b) average stream flow ($Q_{avg}$), (c) average groundwater table ($Gw_{avg}$), and (d) groundwater table range ($\Delta Gw$) for the 5 large storm events included in the former version of the manuscript ("large storms") and the 11 new medium-size events ("medium storms").*

Importantly, these differences in climatic and, especially, hydrological chatacteristics between the large and medium storm events led to marked differences in the resulting stream chemistry: both DOC and $NO_3^-$ concentrations were substantially lower during the medium storm events than during the large storm events (Figure R2 below). This observed pattern is likely caused by the hydrological activation of a thicker and shallower riparian layer during large storm events that leads to the mobilization of relatively larger amounts of DOC and $NO_3^-$ stored in the riparian soil compared to the amounts mobilized from the deeper and narrower layers that are hydrologically activated during medium storm events. These results suggest that large and medium storm events display a distinct mechanism of DOC and $NO_3^-$ mobilization from the riparian zone that precludes a direct integration of the samples from the two type of events into the PLS regression analysis and into the conceptual model we propose in Figure 7 of the manuscript.

[Figure]

[Figure]

*Figure R2. Box plots of dissolved organic carbon (DOC) concentrations and nitrate (NO₃⁻) concentrations in stream waer for the 5 large storm events included in the former version of the manuscript ("large storms") and the 11 new medium-size events ("medium storms").*

[Figure]

*Figure R3. Biplot of a two-component partial least square (PLS) regression model with score and loading vectors re-scaled into the -1 to +1 numerical range in order to display the relative relationships between observations (i.e. 5 large storm events versus 11 medium storm events), predictors, and response variables.*

To illustrate our point further, we have also reanalyzed the PLS regression model including both the former large (N = 5, denoted with #) and the new medium (N = 11, denoted with *) storm events (Figure R3 above). The relative ordination of the predictors is similar to the original PLS regression model, with *P-30*, $Q_{avg}$, and $Q_{avg}/Gw_{avg}$ located relatively contigous in one side of the ordination and *slope* located in the opposite side. However, compared to the original analysis, the model goodness of fit is reduced ($R^2Y$ decreased from 0.96 to 0.72 for DOC and from 0.94 to 0.55 for $NO_3^-$) and the model predictive ability is compromised ($Q^2Y$ decreased from 0.88 to 0.49 for DOC and from 0.82 to 0.25 for $NO_3^-$). Remarkably, the two type of storm events fall in two opposite regions of the biplot: medium storm events cluster in the right side of the ordination, while large storm events make a broader cluster in the left side of the ordination. This result further demonstrate the different nature of the two type of events and the different implications for DOC and $NO_3^-$ mobilization.

All in all, in the revised manuscript we will keep the five large storm events in our PLS regression model, which is the basis for our suggested conceptual model that only applies to such large events. Nevertheless, we will also underscore the hydrological differences between large and medium storm events and discuss the biogeochemical implications of such differences in terms of DOC and $NO_3^-$ mobilization and concentrations in the stream. For that, we will integrate in the main body and the supplementary material of the revised manuscript a condensed version of the information we have presented in this document in this regard.

Adding data beyond 2010-2012

Unfortunately, we do not have chemistry data at the temporal (i.e. daily) or spatial (i.e. three sites along the stream) resolutions that we used in the present study beyond the period that we have analysed here (i.e. 2010-2012). Therefore, there is no other data outside this period that can be sensibly integrated into our analyses and we only use other studies on the catchment to support our discussion.

Ammonium and DON concentrations and other carbon to nitrogen ratios

You have made the right point, $NO_3^-$ is the overwhelmingly dominant form of both inorganic and total nitrogen in our system and thus the other forms lack relevance in the context of the story. For the study period (2010-2012), we have data on daily concentrations of other forms of dissolved inorganic nitrogen (DIN), including $NH_4^+$ and $NO_2^-$ (partially published in Lupon et al., 2016b). Overall average stream concentrations of $NH_4^+$ (0.01 ± 0.006 mg N $L^{-1}$) and $NO_2^-$ (0.006 ± 0.005 mg N $L^{-1}$) are significantly lower than overall average stream $NO_3^-$ concentrations (0.20 ± 0.09 mg N $L^{-1}$). Moreover, $NH_4^+$ and $NO_2^-$ concentrations are in all cases lower than 0.02 mg N $L^{-1}$ and show no differences between baseflow and storm flow conditions. Hence, $NO_3^-$ accounted for more than 90% of DIN during both base flow and storm flow during the study period. It its turn, dissolved organic nitrogen (DON) was always below 0.05 mg mg N $L^{-1}$, which implies that $NO_3^-$ makes up more than 80% of the total dissolved nitrogen under all circumstances. We are therefore confident that $NO_3^-$ is the major source of nitrogen for stream heterotrophic microorganisms in our system and decided to only analyze spatiotemporal patterns in $NO_3^-$ concentrations. To support this choice, we will now make clear that $NO_3^-$ is the dominant form of nitrogen in Font del Regàs in the revised manuscript.

*Minor concerns*

*Lines 25-26: The authors state "These results suggest that (i) increased supply of limited resources during storms can promote in-stream heterotrophic activity during high flows . . ." They don't actually measure this process, instead they infer it from DOC:NO3- ratios. Perhaps reword this result to be more accurate.*

**[Reply]**: It is true that we do not explicitly measure heterotrophic activity, but we do provide data and analyses that support an "increased supply of limited resources during storms", which is where most of the weight of this statement lies on. The subsequent implication that this increased supply can promote in-stream heterotrophic activity is indeed inferred from the analyses on DOC:NO$_3^-$ molar ratios and, more importantly, from the increase in DOC and NO$_3^-$ concentrations during storm flow compared to base flow. We will tone down the statement by using "could sustain" instead of "can promote", which is more accurate.

*Lines 108-110: "All data and analyses were integrated and carried out for daily resolutions, which were determined by the availability of the stream chemical data." This line is confusing, please reword.*

**[Reply]**: We agree, and in the revised manuscript we will change this sentence to "All data and analyses were integrated and carried out for daily resolutions, which was the resolution of the stream chemical data".

*Line 114: Please avoid use of "fortnightly" as most readers will not know what this means. Instead state the actual frequency (i.e., every two weeks).*

**[Reply]**: This is most likely true and so we will remove the word "fortnightly" and use "every two weeks" instead. Thanks.

---

## Author Response (AR1)

**Response to Decision on manuscript hess-2021-401**

Hydrology and riparian forests drive carbon and nitrogen supply and DOC:NO3- stoichiometry along a headwater Mediterranean stream

**Editor comments to the authors**

**Dear authors,**

First, I would like to apologize for how long the review process has been since your initial submission. One of the reviewers became unresponsive after their deadline to submit their review had passed, and it was difficult to find another reviewer to replace them. I thank you very much for your patience through all of this. Two referees have now provided comments on your manuscript and suggested ways of strengthening it: while they both found it to be of good quality, they raised that in its current state, its contribution (in terms of novelty) may be modest. Your responses, posted on the Interactive Discussion page, indicate that you have already started addressing some of these comments. I am therefore returning your manuscript for moderate revisions, and I look forward to receiving your revised manuscript. With best regards,

Genevieve Ali

**[Reply]:**

Dear Editor,

We appreciate your apology about the length of the review process. While it is indeed not optimal for us authors to experience long waiting times in the process, we understand editorial work is becoming increasingly challenging, particularly finding suitable reviewers, and more so in the last couple of years due to the well-known circumstances.

We are happy that both reviewers found our work of good quality and we noticed the common suggestions they had, which we have incorporated into the revised manuscript or rebut accordingly.

Please, find below our point-by-point response to the reviews in line with the comments we posted in the online Interactive Discussion, and including descriptions of the relevant changes made to the original manuscript. Note that the text in *italics refers to literal comments by the reviewers*, the text in blue contains literal quotations from the revised version of the manuscript, and line numbers refer to those in the revised, clean version.

With thanks in advance for your effort handling our manuscript,

José L. J. Ledesma and co-authors

**Reviewer #1**

**General comments**

This study analyses DOC, NO3- and the DOC:NO3- ratio along three stations in a Mediterranean headwater, and across flow conditions, during a 2-year period. Spatial variability is controlled by the presence of riparian forest and topography, while temporal variability is controlled by hydroclimatic conditions. The authors conclude that this spatiotemporal variability influences stream metabolic processes.

The manuscript is well written and the conclusions are clear. This work is within the scope of HESS but I find this is a modest contribution to the literature.

Here are three options to make a stronger paper:

- Relax the selection criteria for the storm events to include in the PLS regression (currently only 5 observations)
- Include data on DOC and N composition, not only in the discussion. The discussion suggests that such data is available. Is nitrate the only N form in this stream?
- Include more recent data to make a more complete synthesis of research in this catchment. The references indicate that other studies have taken place in this catchment since the monitoring period 2010-2012 considered here.

**[Reply]**: We are happy that the reviewer found the manuscript "well written" and the conclusions "clear" and appreciate the three suggestions to make the paper stronger. They are sensible and, in fact, we had already considered these options during the preparation of the manuscript. Below we respond to each of these suggestions separately and explain the actions that we have taken in each case.

**Including more storm events in the PLS regression**

Following the reviewer's suggestion, we relaxed the selection criteria for including storm events in our analyses. Specifically, events were included if they fulfilled three requirements: (i) a precipitation amount of at least 25 mm, as opposed to the 50 mm threshold that we had in the former version of the manuscript (note that events below the 25 mm threshold showed no groundwater table response at the temporal resolution of the study and limited stream flow responses), (ii) stream flow was classified as storm flow during the days of the event (not all events with precipitation amounts higher than 25 mm generated storm flow at the temporal resolution of the study), and (iii) a complete stream chemistry data series associated with the event for the downstream site (i.e. no gaps in chemical data for the event dates were allowed as we were ultimately interested in stream chemical responses).

Using these more relaxed criteria, we identified a total of 11 new events besides the 5 events that we had previously analyzed. We calculated the same nine hydroclimatic descriptors defining those 5 former events for the 11 new events, as well as their average DOC and  $NO_3^-$  concentrations and the relative increase in DOC and  $NO_3^-$  concentrations with respect to base flow conditions (Table R1).

**Table R1**. Hydroclimatic descriptors of the storm events identified during the study period (September 2010 to August 2012) at the downstream site of the Font del Regàs catchment, including duration, precipitation amount (P), accumulated precipitation seven (P-7) and 30 (P-30) days before the event, average stream flow ( $Q_{avg}$ ), average groundwater table ( $Gw_{avg}$ ), groundwater table range ( $\Delta Gw$ ), slope of the linear relationship between riparian groundwater table and stream flow (slope), and average stream flow normalized to average groundwater table ( $Q_{avg}/Gw_{avg}$ ). Average dissolved organic carbon (DOC) concentrations and average nitrate ( $NO_3^-$ ) concentrations during each event at the downstream site and the relative increase in DOC and  $NO_3^-$  concentrations with respect to average concentrations observed during base flow conditions (within brackets) are also shown. Events are subdivided into medium (\*) and large (#) storms.

| Even | Period           | Duration
(days) | P
(mm) | P-7
(mm) | P-30
(mm) | Q avg
(mm) | Gw avg
(m) | $\Delta Gw$ (m) | Slope
(m mm -1 ) | $\begin{array}{c} Q_{avg}\!/Gw_{avg} \\ (mm \; m^{\text{-}1}) \end{array}$ | DOC
(mg C L -1 ) | NO3 -
(mg N L -1 ) |
|------|------------------|--------------------|-----------|-------------|--------------|--------------------------|--------------------------|-----------------|--------------------------------|----------------------------------------------------------------------------|--------------------------------|---------------------------------------------|
| *1   | 16-19/09/2010    | 4                  | 52        | 9           | 36           | 0.4                      | 0.97                     | 0.04            | 0.10                           | 0.4                                                                        | 1.01 (15%)                     | 0.23 (36%)                                  |
| *2   | 14-17/05/2011    | 4                  | 30        | 22          | 122          | 0.7                      | 0.90                     | 0.02            | 0.15                           | 0.8                                                                        | 0.99 (13%)                     | 0.17 (5%)                                   |
| *3   | 30/05-02/06/2011 | 4                  | 36        | 0           | 69           | 0.6                      | 0.92                     | 0.02            | 0.15                           | 0.6                                                                        | 1.19 (36%)                     | 0.18 (7%)                                   |
| *4   | 03-06/06/2011    | 4                  | 50        | 36          | 96           | 0.6                      | 0.92                     | 0.03            | 0.14                           | 0.7                                                                        | 0.83 (-5%)                     | 0.17 (-1%)                                  |
| *5   | 09-15/06/2011    | 7                  | 34        | 66          | 148          | 0.6                      | 0.91                     | 0.03            | 0.15                           | 0.7                                                                        | 1.10 (26%)                     | 0.18 (7%)                                   |
| *6   | 25/07-03/08/2011 | 10                 | 74        | 4           | 48           | 0.5                      | 0.94                     | 0.04            | 0.16                           | 0.5                                                                        | 0.82 (-7%)                     | 0.18 (5%)                                   |
| *7   | 23-30/10/2011    | 8                  | 98        | 17          | 39           | 0.3                      | 0.95                     | 0.09            | 0.17                           | 0.3                                                                        | 1.53 (75%)                     | 0.23 (41%)                                  |
| *8   | 21-23/03/2012    | 3                  | 57        | 6           | 8            | 0.8                      | 0.88                     | 0.06            | 0.22                           | 0.9                                                                        | 0.95 (8%)                      | 0.36 (118%)                                 |
| *9   | 03-05/04/2012    | 3                  | 29        | 0           | 63           | 0.7                      | 0.95                     | 0.06            | 0.13                           | 0.7                                                                        | 1.23 (40%)                     | 0.16 (-2%)                                  |
| *10  | 18-24/05/2012    | 7                  | 49        | 10          | 62           | 0.5                      | 0.97                     | 0.04            | 0.02                           | 0.5                                                                        | 0.82 (-7%)                     | 0.19 (12%)                                  |
| *11  | 04-09/08/2012    | 6                  | 33        | 0           | 4            | 0.2                      | 1.05                     | 0.03            | 0.23                           | 0.2                                                                        | 1.08 (23%)                     | 0.18 (5%)                                   |
| #1   | 11-14/03/2011    | 4                  | 95        | 11          | 39           | 1.7                      | 0.74                     | 0.36            | 0.16                           | 2.3                                                                        | 2.35 (167%)                    | 0.40 (140%)                                 |
| #2   | 14-23/03/2011    | 10                 | 116       | 102         | 133          | 3.3                      | 0.64                     | 0.55            | 0.14                           | 5.1                                                                        | 2.16 (146%)                    | 0.33 (97%)                                  |
| #3   | 02-12/11/2011    | 11                 | 134       | 65          | 116          | 0.8                      | 0.87                     | 0.21            | 0.18                           | 0.9                                                                        | 2.45 (178%)                    | 0.40 (138%)                                 |
| #4   | 14-19/11/2011    | 6                  | 174       | 14          | 261          | 4.2                      | 0.69                     | 0.52            | 0.06                           | 6.1                                                                        | 1.91 (117%)                    | 0.29 (74%)                                  |
| #5   | 21-30/11/2011    | 10                 | 67        | 182         | 426          | 3.4                      | 0.74                     | 0.21            | 0.07                           | 4.5                                                                        | 1.36 (55%)                     | 0.34 (102%)                                 |

We observed large differences between the two subsets of events (the new and the former) in their precipitation and associated hydrological response and consequently classified them as 'medium storms' and 'large storms'. Compared to the large storms, the medium storm events were characterized by significantly (a) lower accumulated precipitation amounts, (b) lower average stream flow, (c) deeper groundwater tables, (d) smaller groundwater table ranges (i.e. the thickness of the riparian layer that becomes hydrologically activated during the event), (e) lower stream DOC concentrations, and (f) lower stream NO3- concentrations (Wilcoxon rank-sum tests, p<0.01 in all cases; Figure R1). This analysis suggest that the differences in climatic and, especially, hydrological chatacteristics between the large and medium storm events were responsible for the marked differences in the resulting stream DOC and NO3- concentrations. Sepecifically, we argue that the hydrological activation of a thicker and shallower riparian layer during large storm events leads to the mobilization of relatively larger amounts of DOC and NO3- stored in the riparian soil compared to the amounts mobilized from the deeper and narrower layers that are hydrologically activated during medium storm events. We conclude that large and medium storm events display a distinct mechanism of DOC and NO3- mobilization from the riparian zone, and thus including the samples from the two type of events into the same PLS regression analysis

would not help shedding light on the hydroclimatic characteristics that most effectively promote the mobilization of DOC and NO3- into the stream via groundwater table elevation and consequent hydrological activation of upper riparian layers, as highlighted in the conceptual model we propose in Figure 7 (now Figure 8) of the manuscript.

**Figure R1**. Box plots of (a) precipitation amount (P), (b) average stream flow ( $Q_{avg}$ ), (c) average groundwater table ( $Gw_{avg}$ , with 0 value indicating the soil surface and negative values indicating cm below the soil surface), (d) groundwater table range ( $\Delta Gw$ ), (e) average dissolved organic carbon (DOC) concentrations at the downstream site, and (f) average nitrate ( $NO_3^-$ ) concentrations at the downstream site for 'large' (N = 5) and 'medium' (N = 11) storm events identified during the study period (September 2010 to August 2012). On each box, the central mark indicates the median, and the bottom and top edges of the box indicate the 25th and 75th percentiles, respectively. The whiskers extend to the most extreme data points not considered outliers. Results from pairwise Wilcoxon rank-sum tests are also shown, where \* indicates statistical difference (p<0.01).

All in all, the inclusion of medium storm events is an added value to the manuscript because the new results underscore the hydrological and biogeochemical differences between the two types of events. Consequently, we have decided to keep only the five large storm events in our PLS regression model, which is the statistical analysis that supports our suggested conceptual model, and which only applies to large events where clear hydrological activation of upper riparian soil layers takes place. Hence, in the revised version of the manuscript, we have included Table R1 and Figure R1 showed here (as new Table A1 and new Figure 5, respectively), and integrated in the text all information regarding these new analyses, including additions in the Abstract (Lines 21-23: "The hydroclimatic analysis of storms suggest that large and medium storm events display a distinct mechanism of DOC and NO3- mobilization. In comparison to large storms, medium storm events showed limited hydrological responses that led to significantly lower stream DOC and NO3- concentrations"); in the Materials and Methods (Lines 180-182: "We further aimed to explore the hydroclimatic characteristics that most effectively promoted the mobilization of DOC and NO3- into the stream via groundwater table elevation and consequent

hydrological activation of upper riparian layers during storm events"; Lines 184-186: "We then related those descriptors with the observed stream DOC and  $NO_3^-$  concentrations through a partial least square (PLS) regression model for the subset of storms for which clear hydrological activation of upper riparian soil layers occurred"; Lines 188-193: "We identified storm events based on three requirements: (i) a precipitation amount during the days included in the event of at least 25 mm (events below these threshold showed marginal groundwater table responses at the temporal resolution of the study), (ii) stream flow was classified as storm flow during the days of the event (not all events with precipitation amounts higher than 25 mm generated storm flow at the temporal resolution of the study), and (iii) a complete stream chemistry data series associated with the event for the downstream site (i.e. no gaps in chemical data for the event dates were allowed as we were ultimately interested in stream chemical responses)"; Lines 214-220: "We observed large differences between two subsets of events in their groundwater table elevation (defined by their groundwater table range  $\Delta Gw$ ), generally associated with their precipitation amount P ( $R^2 = 0.67$ , linear regression between P and  $\Delta Gw$ ). We classified these two subsets of events as 'medium storms' (P = 29 to 98 mm,  $\Delta Gw = 2$  to 9 cm, N = 11) and 'large storms' (P =67 to 174 mm,  $\Delta Gw$  = 21 to 55 cm, N = 5), and proceeded with the PLS analyses only with the subset of large storms because, unlike medium storms, they showed clear hydrological activation of upper riparian soil layers defined by  $\Delta Gw$ . Pairwise Wilcoxon rank-sum tests were used to compare the hydroclimatic descriptors and the stream DOC and NO3- concentrations between medium and large storm events"); in the Results (Lines 296-301: "Hydrological responses were considerably different between medium and large storm events. Compared to large storms, medium storms were characterized by (i) lower accumulated precipitation amounts, (ii) lower average stream flow, (iii) deeper groundwater tables, and (iv) smaller groundwater table ranges, i.e. smaller thickness of the riparian layer that becomes hydrologically activated during the event [...]. Additionally, both stream DOC and stream NO3- concentrations were significantly lower during the medium storm events than during the large storm events"); and in the Discussion (Lines 436-446: "Based on their hydrological response to precipitation, we were able to clearly identify two subsets of events. Large storms, characterized by higher precipitation amounts, produced considerable stream flow and groundwater table responses, whereas medium storms produced only moderate responses in stream flow and limited groundwater table elevations (Fig. 5). These differences in climatic and, especially, hydrological characteristics between large and medium storm events led to marked differences in the resulting stream chemistry, with both DOC and  $NO_3^-$  concentrations being significantly higher during the large storm events. This observation is likely caused by the hydrological activation of a thicker and shallower riparian layer during large storm events that leads to the mobilization of relatively larger amounts of DOC and NO3- stored in the riparian soil compared to the amounts mobilized from the deeper and narrower layers that are hydrologically activated during medium storm events. These results suggest that large and medium storm events display distinct mechanisms of DOC and  $NO_3^-$  mobilization from the riparian zone and that large storms are responsible for a disproportionally larger supply of these solutes").

**DOM composition and other forms of nitrogen**

Unfortunately, we do not have data on DOM composition at daily resolution or during storm flow conditions, which prevents us for directly integrating DOM composition data into the analyses of the

present study. However, during the same study period (2010-2012), we performed longitudinal surveys of DOM composition on 11 occasions during base flow conditions. The data from these surveys showed that DOM in Font del Regàs has a prominent protein-like character in both riparian groundwater and stream water (Bernal et al. 2018). In the present manuscript, we use this information by referring to the published study in order to support our idea that in-stream heterotrophic activity could be partially sustained during storm flow conditions. We have reworded the sentence in the discussion where we included this information to make clear that the data we have on DOM composition was collected during base flow and that we assume the character is maintained across flow conditions (Lines 417-420: "Further, Bernal et al. (2018) showed that DOM at Font del Regàs has a prominent protein-like character in both riparian groundwater and stream water during base flow conditions, which could favour rapid assimilation even during periods of short water residence times assuming DOM molecular composition maintains part or most of its labile character across flow conditions").

Regarding the different forms of nitrogen,  $NO_3^{-1}$  is not the only form found at the Font del Regàs stream, but it makes up the overwhelming majority of both inorganic and total nitrogen. For the study period (2010-2012), we do have data on daily concentrations of other forms of dissolved inorganic nitrogen (DIN), including NH4+ and NO2- (partially published in Bernal et al., 2015). Overall average stream concentrations of NH4+ (0.010 ± 0.006 mg N L-1) and NO2- (0.006 ± 0.005 mg N L-1) are more than one order of magnitude lower than overall average stream  $NO_3^-$  concentrations (0.20 ± 0.09 mg N L-1). Moreover,  $NH_4^+$  and  $NO_2^-$  concentrations are in all cases lower than 0.02 mg N L-1 and show no differences between base flow and storm flow conditions. Hence, NO3- accounted for more than 90% of DIN during both base flow and storm flow during the study period. It its turn, dissolved organic nitrogen (DON) was always below 0.05 mg N L-1, which implies that NO3- makes up more than 80% of the total dissolved nitrogen under all circumstances. Given that (i) NO3 is the major source of nitrogen for stream heterotrophic microorganisms in our system, and (ii) no significant differences were observed in other DIN forms between base flow and storm flow conditions, we decided to only analyze spatiotemporal patterns in NO3- concentrations. We have added this information in the revised manuscript in Lines 75-**76** ("In this stream, NO3- is the major source of N for stream heterotrophic microorganisms, accounting for more than 90% of the dissolved inorganic N (Bernal et al., 2015) and for more than 80% of the total dissolved N (unpublished data)").

**Including more recent data**

Unfortunately, we do not have chemistry data beyond the studied period at the temporal (i.e. daily) or spatial (i.e. three sites along the stream) resolutions that we used in the present study. Some of the studies from the catchment that we cite, and that the reviewer refers to, did use parts of the dataset that we analyse here, though to answer different questions (e.g. Lupon et al., 2016b; Ledesma et al., 2021). In any case, these studies did not contain data beyond 2012. In some cases, the studies used data from specific experiments or campaigns that cannot be directly incorporated into our analyses. This is the case of the aforementioned Bernal et al. (2018) study, which was based on longitudinal surveys of DOM composition. In autumn 2018, we restarted the hydrological monitoring at the downstream site. Since then, DOC and NO3- concentrations (and consequently DOC:NO3- molar ratios), have only been

measured sporadically. We are planning to conduct a more regular sampling of stream water at this site, but unfortunately, no additional data is available at the moment.

**Specific comments**

Figure 1: add location of the weather station

**[Reply]**: We thank the reviewer for pointing this out. We have added the location of the automatic weather station in a revised version of Figure 1.

L115 "Rating curves obtained from the relationships between stream flow and stream water level measurements were used to construct daily time series of stream flow data at each site" can you provide the rating curves in SI?

**[Reply]**: The rating curves were presented in the supplementary material of Lupon et al. (2016b), published also in *HESS*. We can provide them as a new figure in the Appendix of the present study if this is adequate, but we believe it is redundant to reproduce the same figure in two different papers.

L122 "the dynamics of this dataset capture well the dynamics of the groundwater table variation in the surrounding riparian area and therefore we are confident that the recorded pattern at the monitoring location was representative of the groundwater table variations in the riparian zone" please provide stronger evidence that this piezometer is representative of the whole downstream area.

**Figure R2**. Comparison of pressure transducer versus manually measured groundwater tables. Measurements from each piezometer are presented in a different colour and dotted lines are linear regressions between the two variables for each corresponding colour-coded case (p<0.01 in all cases).

**[Reply]**: The evidence for this statement was presented in Ledesma et al. (2021). In addition, to the groundwater tables presented in this study (recorded at 15 min intervals using a water pressure transducer installed in a piezometer placed 2.5 m away from the stream channel), we also measured groundwater tables manually every two weeks during the same period at seven equidistant (ca. 3 m)

piezometers placed ca. 2 m from the stream channel along the same area where the pressure transducer was located. We compared a total of 45 supplementary groundwater table measurements available at each of the seven piezometers with the data from the pressure transducer, and showed that groundwater table dynamics were notably similar in all cases, which supports the use of the pressure transducer data as representative of the downstream riparian areas in our study (Figure R2). In the revised manuscript, we have rephrased the sentence to make the evidence more explicit in Lines 125-129 ("In a previous study, we showed that the dynamics of the pressure transducer dataset captured well the dynamics of manually-measured groundwater table variations in seven other piezometers located in the surrounding riparian area (Ledesma et al., 2021). Therefore, we are confident that the recorded pattern at the pressure transducer location was representative of the groundwater table variations in the riparian zone soils in the lower parts of the catchment").

L174 "hydroclimatic analysis of large storm events" I understand that the authors chose to analyze the largest storm events because they probably exhibit the clearest signal, but the selection criteria here are very strict and only 5 storm events were kept for analysis. This is a very small number, even though PLS regression can handle datasets with few observations and many variables. Wouldn't it be more interesting to relax the selection criteria and include more storm events?

[Reply]: Please, see our detailed response to this issue above.

L322 "given that this is a predominantly heterotrophic system (Lupon et al., 2016c)." please explain how this was determined (most readers won't read the reference)

**[Reply]**: The stream is predominantly heterotrophic because daily rates of ecosystem respiration  $(5.0 - 10.0 \text{ g } \text{O}_2 \text{ m}^{-2} \text{ day}^{-1})$  are between 10- and 100-fold higher than daily rates of gross primary production  $(0.1 - 0.7 \text{ g } \text{O}_2 \text{ m}^{-2} \text{ day}^{-1})$ , as we measured and reported in Lupon et al. (2016c). Following the suggestion, we have included this information in the revised manuscript in Lines 342-343 ("[...] given that this is a predominantly heterotrophic system where ecosystem respiration rates are between 10- and 100-fold higher than gross primary production rates (Lupon et al., 2016c)").

L340 "This result is in line with the idea that headwater streams can remove substantial amounts of NO3 - within relatively short distances (Peterson et al., 2001) [...] providing groundwater inputs with low NO3 - concentrations driven by denitrification, as observed in temperate forest catchments (Cirmo and McDonnell, 1997)." Both instream removal and dilution from the middle part of the catchment can explain this decrease. Is it possible to estimate the share of each process?

**[Reply]**: In general, the data presented in this study cannot be analyzed and evaluated in a way that would allow estimating the relative contribution of these two processes to the NO3- decrease between the upstream and midstream sites. Nevertheless, in this case we are confident than *in-stream removal* dominates over *dilution from riparian denitrification* because denitrification is low at the riparian soils at Font del Regàs (< 3 µg N kg-1 day-1; Poblador et al., 2017). In fact, they rather sustain large nitrification rates (1 – 2 mg N kg-1 day-1; Lupon et al., 2016a). Yet, even if *in-stream removal* likely accounts for an important fraction of the observed NO3-- decrease, we consider that this process alone cannot explain

the entire  $NO_3^-$  reduction between the upstream and midstream sites. We have suggested potential mechanisms for the comparatively high  $NO_3^-$  concentrations in the upstream site that could add to explain the upstream-midstream pattern (Lines 365-369: "[...] it is unclear why the upstream site showed such consistently higher  $NO_3^-$  concentrations with respect to the midstream site. A combination of steeper slopes (which also favour oxygenation and potential nitrification in soil water) and a vegetation dominated by beech forest (which might uptake lower amounts of  $NO_3^-$  than the oak and riparian forests at the midstream section) might partially explain the higher  $NO_3^-$  concentrations upstream and the upstream-midstream pattern (Schiff et al., 2002; Poblador et al., 2019; Simon et al., 2021)").

**L375 "The magnitude of change between flow conditions was different for DOC and NO3 - at the upstream site..." please specify which of DOC or NO3- increases more.**

**[Reply]**: We have specified this information in the revised manuscript in Lines 398-401 ("At the upstream site, the magnitude of change between flow conditions was different between DOC (which on average increased by 66%) and NO3- (which on average increased by 41%), leading to an increase in the frequency of optimal DOC:NO3- stoichiometric conditions for heterotrophic activity during storm flow").

L395 "another study from Font del Regàs showed that DOM has a prominent protein-like character in both riparian groundwater and stream water (Bernal et al., 2018)" suggest to include this data in the analysis (not only just in the discussion) to make a more complete paper. The speciation of DOC and the N species other than nitrate should be analyzed further.

[Reply]: Please, see our detailed responses to these issues above.

**Technical corrections**

L110 "All data and analyses were integrated and carried out for daily resolutions, which were determined by the availability of the stream chemical data." This sentence is unclear

**[Reply]**: We agree, and in the revised manuscript we have changed the sentence to "All data and analyses were integrated and carried out for daily resolutions, which was the resolution of the stream chemical data" (Lines 113-114).

**Reviewer #2**

**General comments**

This manuscript explores hydrological and riparian controls on DOC and NO3- stream chemistry in an oligotrophic Mediterranean stream. Their rationale was to understand how these dynamics may impact in-stream heterotrophic activity. To do this, the authors use data collected at three sites within the catchment collected over a period 2 years (2010-2012). Their results suggest that spatial (upstream to downstream) patterns are explained by riparian geomorphology and forest coverage and that temporal variability results from hydrological variability. The manuscript is well written, figures are clear, and conclusions are straightforward and supported by data. I particularly enjoyed the use of the hydrological metrics to understand controls on lateral exchanges of DOC and NO3- between the riparian zone. That said, the contributions to the literature are modest. I wonder if the authors could strengthen the manuscript by adding data beyond 2010-2012 and include more (smaller) storm events in their PLSR analysis. Large storms are relatively rare and DOC and NO3- transport during smaller, and likely more frequent, storms may influence stream metabolism more than the larger events. Also, I suspect antecedent conditions are even more important for DOC and NO3- transport dynamics during these smaller events. I also wonder if the authors have NH4+ and/or DON concentrations and might consider looking at the DOC:DIN or DOC:TDN ratios. Perhaps NO3- is the dominant form of N and these other N forms won't impact the story much?

**[Reply]**: We are happy that the reviewer found the manuscript "well written" and the conclusions "straightforward and supported by data" and we acknowledge the appreciation of the use of hydrological metrics. Our detail responses to the three main points that we identified in this general comment follow below.

**Including more storm events in the PLS regression**

Please, see our extended response and description of changes made in the manuscript in relation to this issue in our reply to Reviewer #1 above.

**Adding data beyond 2010-2012**

Unfortunately, we do not have chemistry data at the temporal (i.e. daily) or spatial (i.e. three sites along the stream) resolutions that we used in the present study beyond the period that we have analysed (i.e. 2010-2012). Therefore, there is no other data outside this period that can be sensibly integrated into our analyses and we only use other studies from the catchment to support our discussion.

**Ammonium and DON concentrations and other carbon to nitrogen ratios**

The reviewer made the right point,  $NO_3^-$  is the dominant form of both inorganic and total nitrogen in our system and thus, the other forms lack relevance in the context of the story. For the study period (2010-2012), we have data on daily concentrations of other forms of dissolved inorganic nitrogen (DIN), including  $NH_4^+$  and  $NO_2^-$  (partially published in Bernal et al., 2015). During this period, stream

concentrations of NH4+ (0.010 ± 0.006 mg N L-1) and NO2- (0.006 ± 0.005 mg N L-1) were more than one order of magnitude lower than stream NO3- concentrations (0.20 ± 0.09 mg N L-1). Moreover, NH4+ and NO2- concentrations were similar between base flow and storm flow conditions. Hence, NO3- accounted for more than 90% of DIN during both base flow and storm flow. It its turn, dissolved organic nitrogen (DON) was always below 0.05 mg N L-1, which implies that NO3- makes up more than 80% of the total dissolved nitrogen under all circumstances. We are therefore confident that NO3- is the major source of nitrogen for stream heterotrophic microorganisms in our system, and this is why we decided to only analyze spatiotemporal patterns in NO3- is the dominant form of nitrogen in Font del Regàs (Lines 75-76: "In this stream, NO3- is the major source of N for stream heterotrophic microorganic N (Bernal et al., 2015) and for more than 80% of the total dissolved N (unpublished data)").

**Minor concerns**

Lines 25-26: The authors state "These results suggest that (i) increased supply of limited resources during storms can promote in-stream heterotrophic activity during high flows . . ." They don't actually measure this process, instead they infer it from DOC:NO3- ratios. Perhaps reword this result to be more accurate.

**[Reply]**: It is true that we do not explicitly measure heterotrophic activity, but we do provide data and analyses that support an "increased supply of limited resources during storms" (Lines 27-28), which is where most of the weight of this statement lies on. The subsequent implication (that this increased supply can promote in-stream heterotrophic activity) is indeed inferred from the analyses on DOC:NO3- molar ratios and, more importantly, from the increase in DOC and NO3- concentrations during storm flow compared to base flow. We have toned down the statement to be more accurate as suggested by the reviewer by using "could sustain" (Line 28) instead of "can promote".

*Lines 108-110: "All data and analyses were integrated and carried out for daily resolutions, which were determined by the availability of the stream chemical data." This line is confusing, please reword.*

**[Reply]**: We agree, and in the revised manuscript we have changed this sentence to "All data and analyses were integrated and carried out for daily resolutions, which was the resolution of the stream chemical data" (Lines 113-114).

*Line 114: Please avoid use of "fortnightly" as most readers will not know what this means. Instead state the actual frequency (i.e., every two weeks).*

**[Reply]**: This is most likely true and thus we have removed the word "fortnightly" and use "every two weeks" (Line 118) instead.

New references added to the manuscript (marked with \*), references removed from the manuscript (marked with -), and references used in this response letter (not marked)

- \*Beiter, D., Weiler, M., and Blume, T.: Characterising hillslope-stream connectivity with a joint event analysis of stream and groundwater levels, Hydrol. Earth Syst. Sci., 24, 5713-5744, https://doi.org/10.5194/hess-24-5713-2020, 2020.
- Bernal, S., Lupon, A., Ribot, M., Sabater, F., and Martí, E.: Riparian and in-stream controls on nutrient concentrations and fluxes in a headwater forested stream, Biogeosciences, 12, 1941-1954, https://doi.org/10.5194/bg-12-1941-2015, 2015.
- Bernal, S., Lupon, A., Catalán, N., Castelar, S., and Martí, E.: Decoupling of dissolved organic matter patterns between stream and riparian groundwater in a headwater forested catchment, Hydrol. Earth Syst. Sci., 22, 1897-1910, https://doi.org/10.5194/hess-22-1897-2018, 2018.
- Ledesma, J. L. J., Ruiz-Pérez, G., Lupon, A., Poblador, S., Futter, M. N., Sabater, F., and Bernal, S.: Future changes in the Dominant Source Layer of riparian lateral water fluxes in a subhumid Mediterranean catchment. J. Hydrol., 595, 126014, https://doi.org/10.1016/j.jhydrol.2021.126014, 2021.
- Lupon, A., Sabater, F., Miñarro, A., and Bernal, S.: Contribution of pulses of soil nitrogen mineralization and nitrification to soil nitrogen availability in three Mediterranean forests, Eur. J. Soil Sci., 67, 303-313, https://doi.org/10.1111/ejss.12344, 2016a.
- Lupon, A., Bernal, S., Poblador, S., Martí, E., and Sabater, F.: The influence of riparian evapotranspiration on stream hydrology and nitrogen retention in a subhumid Mediterranean catchment, Hydrol. Earth Syst. Sci., 20, 3831-3842, https://doi.org/10.5194/hess-20-3831-2016, 2016b.
- Lupon, A., Martí, E., Sabater, F., and Bernal, S.: Green light: gross primary production influences seasonal stream N export by controlling fine-scale N dynamics, Ecology, 97, 133-144, https://doi.org/10.1890/14-2296.1, 2016c.
- Poblador, S., Lupon, A., Sabaté, S., and Sabater, F.: Soil water content drives spatiotemporal patterns of CO2 and N2O emissions from a Mediterranean riparian forest soil, Biogeosciences, 14, 4195-4208, 10.5194/bg-14-4195-2017, 2017.
- \*Poblador, S., Thomas, Z., Rousseau-Gueutin, P., Sabaté, S., and Sabater, F.: Riparian forest transpiration under the current and projected Mediterranean climate: Effects on soil water and nitrate uptake, Ecohydrology, 12, 16, 10.1002/eco.2043, 2019.
- \*Simon, J., Bilela, S., and Rennenberg, H.: Nitrogen uptake capacity of European beech (Fagus sylvatica L.) only partially depends on tree age, Trees-Struct. Funct., 35, 1739-1745, 10.1007/s00468-021-02190-z, 2021.

---

## Author Response (AR2)

**Response to Decision on manuscript hess-2021-401**

Hydrology and riparian forests drive carbon and nitrogen supply and DOC:NO$_3^-$ stoichiometry along a headwater Mediterranean stream

**Editor comments to the authors**

*Dear authors,*

*Two referees have evaluated your revised manuscript. One found that the small number of events included in your analysis is a major weakness but still recommended acceptance. Another reviewer recommended major revisions, and they questioned the exclusion of the medium-sized storms (that is leading to a small number of events considered in the end). Given that both reviewers are raising the same concern, here, I am returning your manuscript for further revision. One of the reviewers suggested that you either need to better articulate why your current approach is novel or adjust your analysis/sample size. I agree with these suggestions, which will only make your manuscript stronger.*

*With best wishes,*

*Genevieve*

**[Reply]**:

Dear Editor,

Thank you once more for handling our manuscript and returning it for further revisions. First, we would like to thank Rémi Dupas for the time evaluating our manuscript and for the recommendation to accept it this time around. Given that there is still one remaining concern, which was common in the former review round and it is still shared by both reviewers, we realized that it was necessary to include the medium storm events more explicitly in our analyses and conceptual model in order to strengthen our study. In short, we now have (i) included a second PLS regression model with all (large and medium) storm events, (ii) added the medium storms in Table 2, (iii) added a third panel in Figure 8 describing the medium storms in our conceptual model of DOC and NO$_3^-$ mobilization during storm events, and (iv) revised materials and methods, results, and discussion accordingly. All in all, we are satisfied with the revisions, which, as you anticipated, have made our manuscript stronger.

Please, find below our detailed response to reviewer #2, including descriptions of changes made to the previous version of the manuscript. Note that the text in *italics refers to literal comments by the reviewer*, the text in blue contains literal quotations from the revised version of the manuscript, and line numbers refer to those in the revised, clean version.

With thanks in advance for your effort handling our manuscript,

José L. J. Ledesma and co-authors

**Reviewer #2**

*Major issue:*

*I'm struggling with the exclusion of the medium-sized storms from the PLSR. And I think the exclusion masks the actual novelty of their (small) dataset. First, the authors classify storm size based on the combination of precipitation totals and GW table response, with an emphasis on the latter, as storms in the total precipitation range of 67-98 mm were classified as either medium or large based on whether they produced a large change in the GW table. The authors then proceed with their PLSR analysis with the 5 storms that produced a substantial GW table response AND include change in GW table as a predictor in their PLSR models, which seems like circular logic and tautological. Further, their subsequent conceptual model, based on this analysis, does not build on our current understanding of how DOC and NO3- are transported from riparian areas to streams during large precipitation events during dry vs. wet watershed conditions. It seems like the authors are aware of this because they site a few key papers, which all suggest that you will see large DOC and NO3- transport if there is enough precipitation when the system is transport-limited (buildup of solutes during dry conditions) vs. when the system is more source-limited during wetter conditions.*

*I think the authors are missing an opportunity in their analysis, though I think it might be difficult with the number of events they have (though they proceeded with 5 events previously). The actual novelty in their dataset is about the interaction between precipitation totals and antecedent watershed dryness/wetness, especially in that 67-98 mm event precipitation range. Specifically, what is the threshold in antecedent watershed dryness/wetness as quantified using P-7/30 when storms actually produce a GW table response and export more DOC and NO3-? We don't typically classify storm size based on the GW table response, we measure and quantify storm size based on precipitation totals. And this is how we make predictions about how storm size influences stream biogeochemistry.*

*I wish I had the analytical solution for including the interaction between total precipitation and P-7/30 and/or finding the threshold in P-7/30 that produces at GW table response, but I don't. Maybe the authors can include interactions in the PLSR? Regardless, for this portion of the manuscript to build on our understanding of DOC and NO3- export during storms, the authors need to include the data from medium-sized storms and adjust their conceptual model to reflect this. There is novelty there, the authors just need to think a little more about this final analysis.*

**[Reply]**: We thank the reviewer for their detailed evaluation of our manuscript and for their perseverance in requesting further clarifications/interpretations in the relation to the storm event analysis. First, we must note that in our replies to the reviewers' comments that were posted in the online Interactive Discussion we included a PLS regression model that contained all (large and medium) storm events. We used that figure to illustrate the different nature of large *versus* medium storm events in relation to DOC and NO$_3^-$ mobilization and to justify the exclusion of the medium storm events from the analyses and conceptual model. However, in order to keep it simple, we did not include this figure in our former response letter, which would have helped to clarify our case. At any rate, we now realize that this PLS regression analysis and subsequent interpretations not only should be included in the

response letter, but also in the manuscript. Therefore, the present revised version includes results from this PLS regression analysis and further information of both large and medium storm events characterized in this study.

Reply to the general concern and description of changes made in the manuscript

In general, the essence of the reviewer comment lies on the fact that we excluded the medium storm events from our PLS regression analysis and from further interpretation in our conceptual model. We have explicitly addressed this concern by including in the manuscript a PLS regression model where all 16 storm events (5 large and 11 medium) act as observations, the nine hydroclimatic descriptors as predictors, and stream DOC and $NO_3^-$ concentrations as response variables. The inclusion of this new PLS regression model have, in fact, helped to make our case stronger in a number of ways. First, in the PLS ordination biplot we observed two clear clusters falling in opposite sides across the first component. These two clusters precisely correspond to large and medium storm events, respectively. Moreover, the hydroclimatic descriptors considered important in this model (VIP>1) were precipitation amount ($P$), groundwater table range ($\Delta Gw$), and average groundwater table ($Gw_{avg}$). These results provide further detail supporting our former classification based on storm size and groundwater table response and the idea that large and medium storm events display a distinct mechanism of DOC and $NO_3^-$ mobilization. Additionally, these results highlight that storm size and associated groundwater table responses provide a first order control (in terms of relevance) on the mobilization of DOC and $NO_3^-$ from the riparian zone to the streams during storm events.

Furthermore, the nine hydroclimatic descriptors showed small variability among the medium storm events, but varied widely among the five large storm events. For this reason, we performed a second PLS regression analysis using only the subset of large storm events (i.e., our original PLS regression model) in order to gain further insights into the hydroclimatic characteristics that most effectively mobilize DOC and $NO_3^-$ during these larger events. Indeed, the results from this second PLS indicate that during large storm events the mobilization of DOC and $NO_3^-$ depends on antecedent soil moisture conditions and the relationship between riparian groundwater tables and stream flow (important descriptors in this second model were *P-30*, $Q_{avg}$, *slope*, and $Q_{avg}/Gw_{avg}$), as explained in the previous versions of the manuscript. In light of these results, we conclude that antecedent soil moisture conditions and the relationship between groundwater tables and stream flow play a minor role during medium storm events in the mobilization of DOC and $NO_3^-$ from the riparian zone, but are essential for understanding such mobilization during large storm events.

We have integrated all these results in the revised version of the manuscript. We have added a new panel in Figure 6 (now Figure 5) showing the PLS ordination biplot of the whole dataset (large + medium storms) together with the former PLS ordination biplot including only the five large storm events. Former Table A1 is now Table 2, where hydroclimatic descriptors and chemical characteristics are shown for both large and medium storm events. Figure 8 conceptualizing the mobilization of DOC and $NO_3^-$ from the riparian zone to the stream during storm events now includes an additional panel for the conceptualization of medium storms.

We have simplified the material and methods to clarify that we use PLS regression models that consider the whole storm events dataset  (Lines 183-184: "We then related those descriptors with the observed stream DOC and $NO_3^-$ concentrations using partial least square (PLS) regression models").

We have now extensively described the results of the new PLS regression model in the results section, as well as presented the classification between large and medium storm events in this part using the results from the new PLS to support it (Lines 289-296: "A two-component PLS regression model using these 16 storm events as observations and the nine hydroclimatic descriptors as predictors explained 72% (i.e., $R^2Y$ = 0.72) of the variation in average stream DOC concentration and 55% (i.e., $R^2Y$ = 0.55) of the variation in average stream $NO_3^-$ concentration. The $Q^2Y$ values, representing the ability of the model to predict new data, were 0.49 for DOC and 0.25 for $NO_3^-$, both below the 0.50 threshold for good models. Three out of the nine predictors reached VIP>1, indicating that they were important for explaining the variability in the response variables (Fig. A2a). Specifically, higher precipitation amount (higher $P$), larger groundwater table range (higher $\Delta Gw$), and shallower average groundwater table (lower $Gw_{avg}$) related to higher stream concentrations of both DOC and $NO_3^-$, which fell close together in the PLS ordination biplot (Fig. 5a)"; Lines 297-310: "Remarkably, we observed two clusters of observations falling in two opposite sides across the first component of the PLS biplot: a dense cluster of eleven storm events located in the right side, close to $Gw_{avg}$ and opposite $P$ and $\Delta Gw$, and a more scattered cluster of five storm events located in the left side, closer to $P$ and $\Delta Gw$ and far from $Gw_{avg}$ (Fig. 5a). Indeed, there were large hydroclimatic differences between these two subsets of events. The eleven storm events clustered in the right side were characterized by (i) lower $P$, (ii) lower average stream flow ($Q_{avg}$), (iii) deeper $Gw_{avg}$, and (iv) smaller $\Delta Gw$ (i.e., smaller thickness of the riparian layer that becomes hydrologically activated during the event) compared to the five storm events located in left side (Wilcoxon rank-sum test, p<0.01 in all cases; Fig. 6a-d). Given the large difference in $P$ and associated $\Delta Gw$ between the two subsets of events, we classified them as 'medium storms' ($P$ = 29 to 98 mm, $\Delta Gw$ = 2 to 9 cm, n = 11) and 'large storms' ($P$ = 67 to 174 mm, $\Delta Gw$ = 21 to 55 cm, n = 5). Additionally, both stream DOC and stream $NO_3^-$ concentrations were significantly lower during the medium storm events than during the large storm events (Wilcoxon rank-sum test, p<0.01 in both cases; Fig. 6e-f). As a result, stream DOC concentration increased (on average) with respect to the average concentration observed during base flow conditions by 133% during large storms and only by 20% during medium storms. Likewise, stream $NO_3^-$ concentration increased 110% during large storms and only 21% during medium storms compared to average base flow conditions"; Lines 324-326: "Given their variability in hydroclimatic descriptors and in stream DOC and $NO_3^-$ concentrations, we performed a second PLS regression analysis using only the subset of five large storms in order to gain further insights into the hydroclimatic characteristics that most effectively mobilize DOC and $NO_3^-$ during these larger events").

Finally, we have extensively included the medium storm events in our interpretations and in the context of our conceptual model in the discussion section (Lines 448-461: "Results from the first PLS regression model including all storm events showed that precipitation amounts and associated hydrological responses clearly distinguish two subsets of events. Large storms, characterized by higher precipitation amounts, produced higher stream flows and considerable groundwater table elevations, whereas

medium storms produced only moderate responses in stream flow and limited groundwater table elevations (Table 2; Fig. 6). These differences in climatic and hydrological characteristics between large and medium storm events led to marked differences in the resulting stream chemistry, with both DOC and $NO_3^-$ concentrations being significantly higher during the large storm events. Hence, the first PLS showed that higher precipitation (higher $P$), larger groundwater table elevations (higher $\Delta Gw$), and shallower groundwater tables (lower $Gw_{avg}$) related to higher stream DOC and $NO_3^-$ concentrations (Fig. 5; Fig. A2). Mechanistically, chemical differences between the two types of events are likely explained by the hydrological activation of a thicker and shallower riparian layer during large storm events that leads to the mobilization of relatively larger amounts of DOC and $NO_3^-$ stored in the riparian soil compared to the amounts mobilized from the deeper and narrower layers that are hydrologically activated during medium storm events (Fig. 8a). These results suggest that large and medium storm events display distinct mechanisms of DOC and $NO_3^-$ mobilization from the riparian zone and that large storms are responsible for a disproportionally larger supply of these solutes"; Lines 462-463: "Thus, storm size and associated groundwater table responses provide a first order control (in terms of relevance) on the mobilization of DOC and $NO_3^-$ from the riparian zone in the Font del Regàs catchment"; Lines 516-525: "Our conceptual model of riparian DOC and $NO_3^-$ mobilization during large storm events based on data from a sub-humid Mediterranean catchment is similar to previous conceptualizations presented for temperate sites. The model highlights that solute mobilization depends on antecedent soil moisture conditions and the relationship between riparian groundwater table and stream flow (Werner et al., 2019; Beiter et al., 2020). Thus, the mechanisms proposed here for large storm events can be representative of other forest headwaters located in similar climatic settings. In addition, our analyses indicate that in Mediterranean forest headwaters, 'medium storms' show limited groundwater table responses and therefore solute mobilization is more restricted and less dependent on antecedent soil moisture conditions or the relationship between riparian groundwater table and stream flow. This distinctive feature of Mediterranean catchments compared to temperate sites might relate to their overall higher temperatures and evapotranspiration rates and, thus, could extent geographically in the future as climate becomes warmer in temperate areas (Spinoni et al., 2018)").

Groundwater table responses and antecedent conditions in the 67-98 mm event precipitation range

The reviewer correctly points out that "*storms in the total precipitation range of 67-98 mm were classified as either medium or large based on whether they produced a large change in the GW table*", and requests us to investigate the threshold for antecedent conditions that lead to significant groundwater table responses in that precipitation range. While this question is interesting, it is both difficult to resolve with the data at hand and somehow outside of the scope of the present work.

Nevertheless, we have addressed this question qualitatively in the discussion section. Our results suggest that seasonality is the key factor determining whether a storm event in the given precipitation range will result in either large or small groundwater table responses. We have also indicated that future studies should further investigate this interesting topic (Lines 463-479: "[…] in our dataset of 16 storms, we observed an overlap in the 67 to 98 mm precipitation range in which two events (*6 and *7) showed small groundwater table elevations (4 and 9 cm, respectively) and were classified as medium storms, and two other events (#1 and #5) showed significant groundwater table elevations (36 and 21

cm, respectively) and were classified as large storms. Therefore, there is an apparent threshold range in which storm size (defined only by precipitation amount) can lead to either substantial or limited groundwater table responses and, thereby, to relatively more or less DOC and $NO_3^-$ mobilization. While the sample size (n = 4) is too small to clearly determine what processes or pre-conditions define whether a storm in the 67 to 98 mm range will lead to significant groundwater table responses, we noted that medium storm events *6 and *7 took place in July-August and October, respectively, whereas large storm events #1 and #5 took place in March and November, respectively. We argue that seasonality and, in particular, the differences in evapotranspiration between the vegetative and the dormant seasons, might provide a plausible explanation in this context. Mediterranean catchments such as Font del Regàs experience long periods of high evapotranspiration that lead to catchment drying and hydrological disconnection in late spring-summer and a subsequent re-wetting in autumn-late autumn (Medici et al., 2008). In this sense, the time of the year when a storm occurs might determine the magnitude of the riparian groundwater table response and the consequent DOC and $NO_3^-$ mobilization for intermediate-medium to large storm sizes in Mediterranean catchments. Future studies should specifically investigate this ambiguous catchment response to similar precipitation inputs in light of the results presented here (e.g. Beiter et al., 2020)").

Concluding remark

Overall, we think that the extended changes described help to highlight and strengthen the novel aspects of our work. We trust that the extended explanations provided above and the exhaustive revisions included in the manuscript offer (i) a more rigorous justification of the classification between large and medium storm events and of the implementation of the PLS analysis that does not appear circular or redundant, (ii) a better explanation of how storm size operates in this context, (iii) a more complete conceptual framework that can be clearly linked to our current understanding of how DOC and $NO_3^-$ are mobilized from riparian zones to streams during storm events, and (iv) clear support from appropriate literature describing the physical mechanisms that we believe to be relevant in explaining our results and conceptual model.

*Minor comments:*

*Lines 274-276: I'm confused about what the numbers in the parentheses indicate here (eg., 33%). Does the 33% indicate that the ratios were optimal 33% of all storm flow time? Please clarify what larger whole these percentages are a part of.*

**[Reply]**: Yes, this is correct; the 33% refers to all storm flow time. The other percentages given in these lines are to be interpreted in the same way. To avoid confusion, we have made this clarification explicit in the text.

*Lines 218-282: Same issue as above. Does 11% mean 11% of the time during baseflow there were N-limited conditions?*

**[Reply]**: Yes. We have now made it explicit.